# Robust Spatio-Temporal Centralized Interaction for OOD Learning

**Jiaming Ma** [1]  **Binwu Wang** [* 1 2]  **Pengkun Wang** [1 2]  **Zhengyang Zhou** [1 2]  **Xu Wang** [1 2]  **Yang Wang** [* 1 2]

## Abstract

Recently, spatio-temporal graph convolutional networks have achieved dominant performance in spatio-temporal prediction tasks. However, most models relying on node-to-node messaging interaction exhibit sensitivity to spatio-temporal shifts, encountering out-of-distribution (OOD) challenges. To address these issues, we introduce **S**patio-**T**emporal **O**OD **P**rocessor (STOP), which employs a centralized messaging mechanism along with a message perturbation mechanism to facilitate robust spatio-temporal interactions. Specifically, the centralized messaging mechanism integrates Context-Aware Units for coarse-grained spatio-temporal feature interactions with nodes, effectively blocking traditional node-to-node messages. We also implement a message perturbation mechanism to disrupt this messaging process, compelling the model to extract generalizable contextual features from generated variant environments. Finally, we customize a spatio-temporal distributionally robust optimization approach that exposes the model to challenging environments, thereby further enhancing its generalization capabilities. Compared with 14 baselines across six datasets, STOP achieves up to **17.01%** improvement in generalization performance and **18.44%** improvement in inductive learning performance. The code is available at https://github.com/PoorOtterBob/STOP.

## 1. Introduction

Spatio-temporal prediction is a critical task in urban computing with positive effects for various applications including traffic management, energy control, and atmospheric analy-

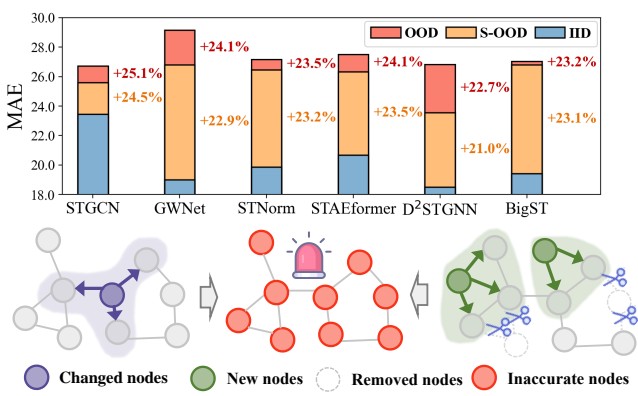

*Figure 1.* Subfigure (a) (upper half) illustrates the predictive performance of STGNNs in various scenarios and subfigure (b) (lower half) demonstrates the sensitivity of the node-to-node messaging mechanism to spatio-temporal shift.

sis (Xia et al., 2024; Liang et al., 2023; Miao et al., 2024; Zhang et al., 2023; Liu et al., 2024a). Currently, spatio-temporal graph convolutional networks (STGNNs) have emerged as the primary tools in this field. These models generalize spatio-temporal graph dependencies and employ node-to-node messaging mechanism (such as GCN or Transformer) for feature interaction. Finally, the generated unique representation is fed into a decoder to generate prediction.

However, the encouraging success of STGNNs is predicated on the independent and identically distributed (IID) assumption. In reality, the distributional characteristics (such as mean and variance) or graph structures of spatio-temporal data evolve over time, presenting out-of-distribution (OOD) generalization challenges for STGNNs.

With the LargeST-SD dataset (Liu et al., 2023b) as an example, we report the performance of advanced STGNNs in both IID and OOD scenarios, as shown in Figure 1 (a). The results indicate that their performance rapidly deteriorates when facing spatio-temporal OOD challenges, particularly in structural shift (S-OOD) scenario. Embarrassingly, the models' reliance on node-to-node messaging appears to hinder their effectiveness. Ablation experiments (as shown in Table 2) further support this: for some advanced STGNNs, variants without node-to-node messaging mechanism per-

[1]University of Science and Technology of China (USTC), Hefei, China [2]Suzhou Institute for Advanced Research, USTC, Suzhou, China. First author mail: JiamingMa@mail.ustc.edu.cn. Correspondence to: Binwu Wang and Yang Wang as corresponding authors <wbw2024@ustc.edu.cn and angyan@ustc.edu.cn>.

*Proceedings of the 42nd International Conference on Machine Learning*, Vancouver, Canada. PMLR 267, 2025. Copyright 2025 by the author(s).

form embarrassingly better. This is because the knowledge learned acquired through this mechanism is coupled with the features of the training graph, and this knowledge is difficult to generalize to unseen graphs during testing (Hamilton et al., 2017). As illustrated in Figure 1 (b), when node features change, STGNNs struggle to accurately represent these nodes. Furthermore, when certain nodes disappear from the graph, their neighbors are significantly impacted, as they can no longer aggregate information along the trained paths. This propagation of errors through the node-to-node messaging mechanism adversely affects the accuracy of the entire graph representation. On the other hand, generating accurate representations for new nodes, i.e., inductive learning, also poses a significant challenge for STGNNs (Zheng et al., 2023; 2024; Wang et al., 2024b).

In this paper, we propose a **S**patio-**T**emporal **O**OD **P**rocessor (STOP). Specifically, STOP's core contribution is a spatio-temporal centralized interaction strategy: first, we design a centralized message passing mechanism configured with Context-Aware Units (ConAU) to learn generalizable contextual features. The interaction is realized through message exchange between ConAU and nodes - blocking node-to-node messages. Furthermore, we employ a Generalized Perturbation Unit (GenPU) to randomly perturb the message interaction process, effectively promoting diverse training environments through the introduction of uncertainty, preventing the model from coupling with a single training environment. Furthermore, we customize a spatio-temporal Distributionally Robust Optimization (DRO) objective for GenPU to help the model learn robust knowledge from challenging environments. STOP can effectively generalize this knowledge to new nodes to generate good representations. Finally, STOP synthesizes two prediction components from temporal and spatial dimensions to generate the final prediction, which can enhance model robustness across comprehensive OOD scenarios.

Our contributions can be four-folds: ❶ We introduce a Spatio-Temporal OOD Processor (STOP), incorporating a robust centralized messaging mechanism and a message perturbation mechanism. ❷ The centralized messaging constrains nodes to interact exclusively with Context Aware Units (ConAU) for feature interaction, thereby enhancing the resilience of model to spatio-temporal shifts. ❸ The message perturbation mechanism, equipped with Generalized Perturbation Units (GenPU), disrupts node interactions with ConAU and includes a specialized spatio-temporal distributionally robust optimization (DRO) for GenPU, facilitating the model's acquisition of causal knowledge across diverse environments. ❹ We evaluate STOP's effectiveness against 14 baselines across six datasets, where it can achieve up to **17.01%** improvement in generalization performance and up to **18.44%** improvement in inductive learning performance.

## 2. Preliminaries

We use a graph $\mathcal{G} = (\mathcal{V}, \mathbf{A})$ to represent spatio-temporal data, where $\mathcal{V}$ means the node set with $N$ nodes and $\mathbf{A} \in \mathbb{R}^{N \times N}$ is the weighted adjacency matrix of the graph $\mathcal{G}$. We use $X_t \in \mathbb{R}^{N \times c}$ to represent the observed graph signal at time step $t$, where $c$ indicates the number of feature channels.

Training environment $e^*$ contains two characteristic elements: a training graph $\mathcal{G}^* = (\mathcal{V}^*, \mathbf{A}^*)$ and training data $(\mathcal{X}^*, \mathcal{Y}^*)$. With this training environment, spatio-temporal OOD learning aims to learn a robust function $f$, which can accurately predict values after $T_P$ time steps given observed data of past $T$ time steps $\mathbf{X} = [X_1, X_2, \ldots, X_T] \in \mathbb{R}^{T \times N \times c}$ and the graph sampled from any environment $e \sim \mathcal{E}$, where $e$ may have different two characteristic elements with training environment $e^*$,

$$\mathrm{argmin}_f \sup_{e \in \mathcal{E}} \mathbb{E}_{(\mathbf{X}, \mathbf{Y}) \sim p(\mathcal{X}, \mathcal{Y}|e)} \left[ \mathcal{L}\left(f\left(\mathbf{X}\right), \mathbf{Y}\right) \right], \quad (1)$$

where $\mathbf{Y} = \left[X_{T+1}, X_{T+2}, \ldots, X_{T+T_p}\right] \in \mathbb{R}^{T_p \times N \times c}$ is the ground-truth value.

## 3. Methdology

STOP employs a channel mixing module to model temporal and spatial dynamics, and after utilizing centralized messaging and message perturbation mechanisms for feature interaction, the generated temporal and spatial prediction components jointly determine the final prediction. The details of STOP are illustrated in Figure 2 and Algorithm 1.

### 3.1. Temporal Prediction Component

**Temporal decomposition.** In time series analysis, researchers (Cleveland et al., 1990; Wu et al., 2021; Zeng et al., 2023) often decompose time series into components at various time scales. Long-term patterns, such as seasonal or periodic trends, are relatively stable, while short-term patterns, like hourly traffic fluctuations, are unstable. Intuitively, when the spatio-temporal distribution of the node changes over time, long-term patterns may remain robust (Wang et al., 2024b). Hence, we employ temporal decomposition technique to model temporal dynamics at different scales. Specifically, we use the padding moving average kernel $\mathrm{AvgPool}\left(\cdot; \xi\right)$ with kernel size $\xi$ to decouple the input $\mathbf{X} \in \mathbb{R}^{T \times N \times c}$ into long-term patterns $\mathbf{X}_l$ and short-term patterns $\mathbf{X}_s$:

$$\mathbf{X}_l = \mathrm{AvgPool}\left(\mathbf{X}; \xi\right) \in \mathbb{R}^{T \times N \times c}, \quad (2)$$

$$\mathbf{X}_s = \mathbf{X} - \mathbf{X}_l \in \mathbb{R}^{T \times N \times c}. \quad (3)$$

where we employ padding operation $\mathrm{AvgPool}$ in $\left(\cdot; \xi\right)$ along temporal dimension, ensuring a consist time length. Subsequently, two distinct $\mathrm{MLP}\left(\cdot\right) : \mathbb{R}^{T \times N \times c} \to \mathbb{R}^{T \times N \times d}$ are

leveraged to model the temporal interdependencies within these kinds of patterns. Finally, the outputs are mixed to yield the data representation,

$$\mathbf{H}_0 = \text{MLP}_1\left(\mathbf{X}_l\right) + \text{MLP}_2\left(\mathbf{X}_s\right) \in \mathbb{R}^{T \times N \times d_0}. \quad (4)$$

**Spatio-temporal Embedding.** To comprehensively capture spatio-temporal dynamics, we employ embedding techniques to encode various prior information that is independent of spatio-temporal data, which facilitates the model's ability to capture generalizable spatio-temporal knowledge. Specifically, we utilize a timestamp-of-day embedding $\mathbf{E}_t \in \mathbb{R}^{N_t \times d_p}$ to capture the periodic dependencies of each temporal step and a day-of-week embedding $\mathbf{E}_d \in \mathbb{R}^{N_d \times d_p}$ to model the periodic patterns at daily intervals, where $N_d = 7$ is the number of days in one week and $N_t$ indicates the number of sampling points in a day. For example, for some PeMS datasets, the data sampling frequency of traffic flow is five minutes, so $N_t$ is set to $60 \times 24/5 = 288$. $d_p$ is the dimension of each embedding.

In addition, we further use the positional embedding $\mathbf{P}$ followed by Transformer (Vaswani et al., 2017) to encode the position of each data point in $\mathbf{X}$. Finally, we integrate temporal prior embedding and data positional embedding to generate the output $\mathbf{Z}_I$ denoted as the input representation:

$$\mathbf{Z}_I = \text{Concat}\left(\mathbf{H}_0 + \mathbf{P}, \mathbf{E}_t, \mathbf{E}_d\right) \in \mathbb{R}^{T \times N \times (d_0 + 2d_p)}. \quad (5)$$

**Channel Mixing Module.** To capture temporal dynamics, we first mix-up the channel and temporal dimensions of the output $\mathbf{Z}_I$ into shape $N \times d_t$, where $d_t = T * (d_0 + 2d_p)$. Subsequently, we use $L$-layer MLP for hybrid modeling. Given the input of $l$-th MLP layer with residual connection technology $\mathbf{Z}_T^{(l)}$, where $\mathbf{Z}_T^{(0)} = \mathbf{Z}_I$, the forward process of $l$-th MLP layer is as follows:

$$\mathbf{Z}_T^{(l+1)} = \text{GELU}\left(\mathbf{Z}_T^{(l)}\mathbf{W}_1^{(l)}\right)\mathbf{W}_2^{(l)} + \mathbf{Z}_T^{(l)} \in \mathbb{R}^{N \times d_t}, \quad (6)$$

where $l \in \{0, 1, ..., L-1\}$ and $\text{GELU}\left(\cdot\right)$ (Hendrycks & Gimpel, 2016) is an activation function. $\mathbf{W}_1^{(l)} \in \mathbb{R}^{d_t \times 4d_t}$ and $\mathbf{W}_2^{(l)} \in \mathbb{R}^{4d_t \times d_t}$ are learnable parameters. After $L$ MLP layers, we get the temporal representation denoted as $\mathbf{Z}_T = \mathbf{Z}_T^{(L)} \in \mathbb{R}^{N \times d_t}$. Finally, we use a linear transformation as decoder to generate a temporal prediction component $\mathbf{Y}_t$ as follows,

$$\mathbf{Y}_t = \mathbf{Z}_T\mathbf{W}_t + \mathbf{b}_t \in \mathbb{R}^{N \times (T_P * c)}, \quad (7)$$

where $\mathbf{W}_t \in \mathbb{R}^{d_t \times (T_P * c)}$ and $\mathbf{b}_t \in \mathbb{R}^{T_P * c}$ are learnable parameters.

### 3.2. Robust Spatio-temporal Centralized Interaction

3.2.1. CENTRALIZED MESSAGING MECHANISM

STGNNs, relying on node-to-node message interaction mechanisms, are sensitive to structural shifts (Finkelshtein et al., 2023; Han et al., 2024b), which limits their generalization capability on unknown graph structures. To address these limitations, we propose a novel centralized message passing mechanism where each graph node interacts specifically with an established context-aware unit through a novel low-rank attention mechanism.

**Context Aware Units.** We first set $K$ context aware units (ConAU), where $K$ is a hyperparameter and $K \ll N$. ConAUs are used to perceive generalizable contextual features from nodes, which nodes extract to achieve interactions. Specifically, we adopt a learnable feature vector $\boldsymbol{c} \in \mathbb{R}^{d_t}$ for each ConAU, where $d_t$ indicates the number of feature channels. Thus, we can get a series of context feature vectors $\mathbf{C} = [\boldsymbol{c}_1, \boldsymbol{c}_2, \ldots, \boldsymbol{c}_K] \in \mathbb{R}^{K \times d_t}$. Next, we propose a multi-head low-rank attention method to achieve the interaction between nodes and ConAUs.

**Multi-head Low-rank Attention.** This mechanism can be summarized in two processes: aggregating node features to extract contextual features and diffusing contextual features for feature interaction between nodes. It takes $\mathbf{Z}_T \in \mathbb{R}^{N \times d_t}$ and $\mathbf{C}$ as input. Inspired by the multi-head mechanism (Vaswani et al., 2017), we utilize distinct linear layers to project Query, Key, and Value separately into $d_h = d_t/h$ dimensions with $h$ heads. Specifically, for the $i$-th head where $i = \{1, 2, \ldots, h\}$, the calculation of low-rank attention is as follows:

$$\mathbf{Z}_c^{(i)} = \mathcal{A}\left(\mathbf{Q}, \mathbf{K}, \mathbf{V}\right) \quad (8)$$

$$= \underbrace{\text{softmax}\left(\alpha\mathbf{Q}\mathbf{K}^\top\right)}_{\textbf{Diffusion}} \times \underbrace{\text{softmax}\left(\alpha\mathbf{K}\mathbf{Q}^\top\right)}_{\textbf{Aggregation}} \mathbf{V}, \quad (9)$$

where $\mathbf{Q} = \mathbf{Z}_T\mathbf{W}_q^{(i)} \in \mathbb{R}^{N \times d_h}, \mathbf{K} = \mathbf{C}\mathbf{J}_{d_t}^{(i)} \in \mathbb{R}^{K \times d_h}$ and $\mathbf{V} = \mathbf{Z}_T\mathbf{J}_{d_t}^{(i)} \in \mathbb{R}^{N \times d_h}$. Here $\alpha$ is a scaling factor and equals to $1/\sqrt{d_h}$. $\mathbf{W}_q^{(i)} \in \mathbb{R}^{d_t \times d_h}$ is a learnable parameter matrix, and $\mathbf{J}_{d_t}^{(i)} \in [0, 1]^{d_t \times d_h}$ is a column submatrix of $d_t$-order identity matrix $\mathbf{I}_{d_t} \in [0, 1]^{d_t \times d_t}$, which contains all rows and the columns $(d_h * (i-1) + 1)$ to $(d_h * i)$ of $\mathbf{I}_{d_t}$. $\mathbf{J}_{d_t}^{(i)}$ is used to project the feature subspace corresponding to the $i$-th head. The computed attention matrix is low-rank with high efficiency, which is explained in Appendix C.1. Finally, we splice outputs of multiple heads to generate representation for nodes: $\mathbf{Z}_c = \text{Concat}\left(\mathbf{Z}_c^{(1)}, \mathbf{Z}_c^{(2)}, \ldots, \mathbf{Z}_c^{(h)}\right) \in \mathbb{R}^{N \times d_t}$.

This attention comprises both aggregation and diffusion processes, as shown in the right half of Figure 2. The aggregation process, denoted by $\mathbf{K}\mathbf{Q}^\top \in \mathbb{R}^{K \times N}$, extracts node features for updating context features. Conversely, the diffusion process, denoted by $\mathbf{Q}\mathbf{K}^\top \in \mathbb{R}^{N \times K}$, disperses the context features to individual nodes to facilitate feature interaction and node representation generation.

**Theory 1 (Low-rank Attention).** Our attention matrix has

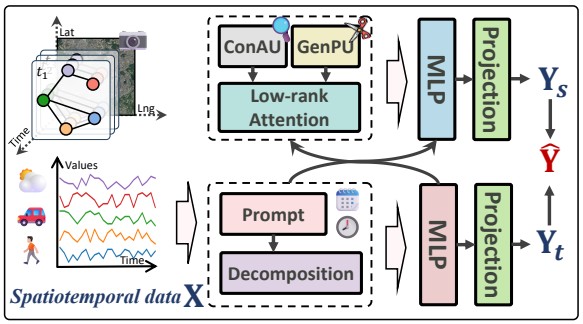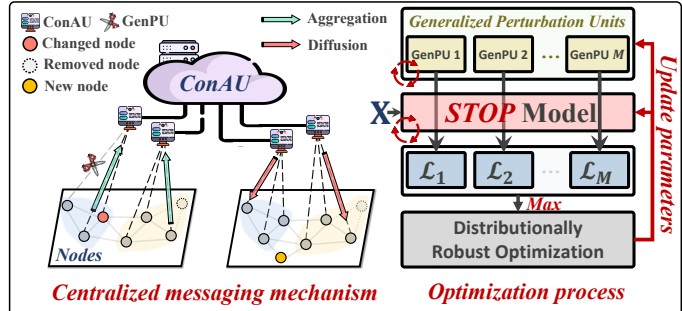

*Figure 2.* Overall architecture (left) and centralized messaging mechanism for robust spatio-temporal interaction (right).

a low-rank property, resulting in high computational efficiency of $\mathcal{O}\left(KNd_h\right)$, which is significantly better than the complexity of vanilla self-attention mechanism $\mathcal{O}\left(N^2 d_h\right)$. Detailed description can be found in Appendix Section C.1.

### 3.2.2. MESSAGE PERTURBATION MECHANISM

We introduce the Generalized Perturbation Units (GenPU) to perturb the interaction process of centralized messaging to improving generalization of the model to unknown environments. Additionally, we specifically design a Distributionally Robust Optimization (DRO) (Duchi & Namkoong, 2019) objective to optimize models and GenPU.

**Generalized Perturbation Units (GenPU).** To acquire robust contextual features, our strategy involves disrupting the aggregation process of the centralized messaging mechanism, which is responsible for updating context features. This approach enables us to circumvent the significant computational overhead associated with directly perturbing the data. Specifically, we create $M$ learnable perturbation vector in the training process, denoted $\mathbb{G} = \{\boldsymbol{g}_1, \boldsymbol{g}_2, \ldots, \boldsymbol{g}_M\}$, where $\boldsymbol{g}_i \in \mathbb{R}^N$ with $i \in \{1, 2, \cdots, M\}$ means $i$-th GenPU. Then, we use $\mathrm{softmax}$ operation to normalize $\boldsymbol{g}_i \in \mathbb{R}^N$ to get the corresponding masking probability vector $\boldsymbol{g}_i' = \mathrm{softmax}\left(\boldsymbol{g}_i\right) \in (0,1)^N$. Subsequently, we create a multinomial distribution $\mathcal{M}\left(\boldsymbol{g}_i'; s\right)$. Based on this distribution, we sample a masking indices $\widetilde{\boldsymbol{g}}_i \sim \mathcal{M}\left(\boldsymbol{g}_i'; s\right) \in \{0,1\}^N$, where $s \in (0, N)$ indicates the number of sample hits (i.e. the number of values equal to 1 in $\widetilde{\boldsymbol{g}}_i$). Finally, we create $K$ replicas of $\widetilde{\boldsymbol{g}}_i$ corresponding to $K$ ConAU. As a result, we can obtain a mask matrix with log operation as follows:

$$\mathbf{G}_i = \log\left(\left[\widetilde{\boldsymbol{g}}_i, \widetilde{\boldsymbol{g}}_i, \ldots, \widetilde{\boldsymbol{g}}_i\right]\right) \in \{-\infty, 0\}^{K \times N}. \quad (10)$$

The logarithmic operation is to facilitate the masking operation during the exponential activation in the subsequent softmax normalization. If $\mathbf{G}_i[m, n] = 0$, the message between $m$-th node and $n$-th ConAU is not be affected. If $\mathbf{G}_i[m, n] = -\infty$, the message between $m$-th node and $n$-th ConAU is masked. Then we integrate $\mathbf{G}_i$ into low-rank

attention mechanism to control the aggregation process:

$$\widetilde{\mathcal{A}}_i\left(\mathbf{Q}, \mathbf{K}, \mathbf{V}; \mathbf{G}_i\right) \quad (11)$$
$$= \mathrm{softmax}\left(\alpha \mathbf{Q}\mathbf{K}^\top\right) \times \underbrace{\mathrm{softmax}\left(\alpha \mathbf{K}\mathbf{Q}^\top + \mathbf{G}_i\right)}_{\textbf{Perturbation operation}} \mathbf{V}.$$

From the perspective of ConAU, random perturbations during the aggregation process introduce environmental uncertainty variations, thereby forcing the model to learn generalizable contextual features from these varying environments. In the training phase, we leverage $M$ GenPU in parallel to conduct the perturbation operation. Accordingly, according to Equation 20, the model will individually generate predictions for these $M$ environments, represented as $\{\widehat{\mathbf{Y}}_1, \widehat{\mathbf{Y}}_2, \ldots, \widehat{\mathbf{Y}}_M\}$, which will be explained below.

**Spatio-temporal Distributionally Robust Optimization.** With $M$ predictions generated from different environments, our propose spatio-temporal DRO, which does not require optimizing all $M$ branches sequentially; instead, it selects the branch with the highest loss for gradient descent, as shown in the right half of Figure 2. This approach indicates that the model performs worst in that particular environment, thereby enhancing training efficiency and encouraging the model to learn purely invariant knowledge. We designate the GenPU responsible for generating this environment as $\boldsymbol{g}$. The specific optimization objective is defined as follows:

$$\min_f \sup_{\boldsymbol{g} \in \mathbb{R}^N} \mathbb{E}_{(\mathbf{X}, \mathbf{Y}) \sim (\mathcal{X}, \mathcal{Y} | e^*)}\left[\mathcal{L}\left(f\left(\mathbf{X}\right), \mathbf{Y}; \boldsymbol{g}\right)\right], \quad (12)$$
$$\text{s.t.} \quad ||\boldsymbol{g}||_0 = s \in (0, N). \quad (13)$$

where $||\cdot||_0$ stands for zero norm. GenPUs participate in the learning process by influencing the sampling distribution of the mask matrix, which is essentially non-differentiable, rather than participating in the backpropagation process as part of the parameters. Thus, we optimize the model parameters and GenPUs alternately, as shown in Algorithm 2.

In fact, our proposed objective belongs to the DRO paradigm, which theoretically has superior generalization

compared to the Empirical Risk Minimization (ERM) paradigm followed by most spatio-temporal models. ERM optimizes the model using only single training environment. The details are provided in Appendix Section D.

**Theory 2.** Within a predefined uncertainty set of environments $\mathcal{E}$, DRO substitutes the expected risk under a single distribution with the worst-case expected risk. DRO yields tighter upper bounds compared to ERM:

$$\sup_{e \in \mathcal{E}} \left\{ \mathbb{E}_{(\mathbf{X}, \mathbf{Y}) \sim p(\mathcal{X}, \mathcal{Y}|e); \mathcal{D}(e, e^*) \leq \rho} \left[ \mathcal{L} \left( f \left( \mathbf{X} \right), \mathbf{Y} \right) \right] \right\}$$
$$\geq \mathbb{E}_{(\mathbf{X}, \mathbf{Y}) \sim p(\mathcal{X}, \mathcal{Y}|e^*)} \left[ \mathcal{L} \left( f \left( \mathbf{X} \right), \mathbf{Y} \right) \right] + \phi, \quad (14)$$

where $\phi = \sqrt{2\rho \operatorname{Var}_{(\mathbf{X}, \mathbf{Y}) \sim p(\mathcal{X}, \mathcal{Y}|e^*)} \left[ \mathcal{L} \left( f \left( \mathbf{X} \right), \mathbf{Y} \right) \right]}$ and the distance $\mathcal{D} \left( \cdot, \cdot \right)$ between any $e \in \mathcal{E}$ and training environment $e^*$ is less than or equal to $\rho$, and "sup" means the supremum. This reveals that DRO allows the model to adapt more robustly to various environments than ERM.

**Robustness Analysis**. Centralized message mechanism restricts interactions to operate between nodes and ConAUs, avoiding the complexities of node-to-node interactions. Structural shift does not significantly disrupt the message passing paths between nodes and ConAUs. Furthermore, GenPU introduces uncertain perturbations during this spatio-temporal interaction process, generating diverse training environments. This strategy prevents the model from over-relying on a single training environment. Our spatio-temporal DRO forces the model to interact with the most challenging instances in the generated environments, which can further enhance the model's robustness. Moreover, the learned context features can flexibly extend to newly added nodes, improving its inductive learning capability.

### 3.2.3. SPATIAL PREDICTION COMPONENT

After feature interaction, we extract shared contextual features from ConAUs for each node. To further enhance the node representation, we refine the personalized features of individual nodes. This refinement involves subtracting the contextual features from the temporal representation to isolate the personalized feature representation of each node, denoted as $\mathbf{Z}_p$:

$$\mathbf{Z}_p = \mathbf{Z}_{\mathrm{T}} - \mathbf{Z}_c \in \mathbb{R}^{N \times d_t}. \quad (15)$$

Subsequently, we concatenate the decoupled context features $\mathbf{Z}_c$ and personalized features $\mathbf{Z}_p$, and then linearly map them back to the initial representation.

$$\mathbf{Z}_t' = \operatorname{GELU} \left( \operatorname{Concat} \left( \mathbf{Z}_p, \mathbf{Z}_c \right) \mathbf{W}_1 \right) \mathbf{W}_2 \in \mathbb{R}^{N \times d_t}, \quad (16)$$

$$\widetilde{\mathbf{Z}}_t = \operatorname{LayerNorm} \left( \mathbf{Z}_t' + \mathbf{Z}_{\mathrm{T}} \right) \in \mathbb{R}^{N \times d_t}, \quad (17)$$

where $\mathbf{W}_1 \in \mathbb{R}^{d_t \times 4d_t}$ and $\mathbf{W}_2 \in \mathbb{R}^{4d_t \times d_t}$ are learnable parameters. We then decouple spatial components by calculating the difference between the input representation $\mathbf{Z}_{\mathrm{I}}$ and

the temporal representation $\widetilde{\mathbf{Z}}_t$, denoted as $\mathbf{Z}_s^{(0)} = \mathbf{Z}_{\mathrm{I}} - \widetilde{\mathbf{Z}}_t$. Next, we utilize a channel mixing module with $L$-layer MLP to capture spatial high-dimensional features, with the final output denoted as the spatial representation $\mathbf{Z}_{\mathrm{S}} = \mathbf{Z}_s^{(L)}$. The forward process of the $l$-th MLP layer is as follows:

$$\mathbf{Z}_s^{(l+1)} = \operatorname{GELU} \left( \mathbf{Z}_s^{(l)} \mathbf{W}_3^{(l)} \right) \mathbf{W}_4^{(l)} + \mathbf{Z}_s^{(l)} \in \mathbb{R}^{N \times d_t}, \quad (18)$$

where $\mathbf{W}_3^{(l)} \in \mathbb{R}^{d_t \times 4d_t}$ and $\mathbf{W}_4^{(l)} \in \mathbb{R}^{4d_t \times d_t}$ are learnable parameters. Finally, same as the temporal part, we also use a linear layer to decode the spatial representation $\mathbf{Z}_s$ to produce a prediction from the spatial component:

$$\mathbf{Y}_s = \mathbf{Z}_{\mathrm{S}} \mathbf{W}_s + \mathbf{b}_s \in \mathbb{R}^{N \times (T_P * c)}, \quad (19)$$

where $\mathbf{W}_s \in \mathbb{R}^{d_t \times (T_P * c)}$ and $\mathbf{b}_s \in \mathbb{R}^{T_P * c}$ are learnable parameters.

### 3.3. Final Prediction

We sum the predictions from the spatial and temporal dimensions to get finial prediction $\widehat{\mathbf{Y}}$ as follow,

$$\widehat{\mathbf{Y}} = \mathbf{Y}_t + \mathbf{Y}_s \in \mathbb{R}^{N \times T_P \times c}. \quad (20)$$

## 4. Experiments

In this section, we conduct a comprehensive evaluation of the proposed model. We will answer the following potential questions in the following subsections one-by-one: **Q.1**. What is the generalization performance of STOP in spatio-temporal OOD scenarios? **Q.2**. Is the proposed centralized messaging mechanism effective? **Q.3**. How sensitive is the model to hyperparameters $M$ and $K$? **Q.4**. Is each component of the model valid for OOD capabilities? **Q.5**. How efficient is the model? **Q.6**. What is the inductive learning ability of STOP for new nodes?

In Appendix, we analyze the effectiveness of STOP in various OOD scenarios (temporal OOD or structural OOD in Section E.5, rapid expansion OOD in Section E.6), and provide visualization examples in Section E.13.

*Table 1.* Summary of the used datasets.

| Dataset | Nodes | Edges | Years |
|---|---|---|---|
| LargeST-SD | 716 | 17,319 | $2017 \sim 2021$ |
| LargeST-GBA | 2,352 | 61,246 | $2017 \sim 2021$ |
| LargeST-GLA | 3,834 | 201,363 | $2017 \sim 2021$ |
| LargeST-CA | 8,600 | 525,888 | $2017 \sim 2021$ |
| PEMSD3-Stream | 655 | 1,577 | $2011 \sim 2017$ |
| KnowAir | 184 | 3,796 | $2015 \sim 2018$ |

### 4.1. Experiment Setting

❶ **Setting.** We set both the input and prediction windows to 12 in traffic prediction and 24 in atmospheric prediction.

Temporal decomposition kernel size $\xi$ is equal to 3 in traffic datasets and 7 in KnowAir. The number of ConAU $K$ is set to $\{8, 24, 32, 64, 8, 4\}$ and the number of GenPU $M$ is equal to $\{3, 3, 3, 3, 2, 4\}$ in six datasets in Table 1. The dimensions of embeddings are set to 64. We use 8 heads in multi-head low-rank attention. We implement all models using PyTorch framework of Python 3.8.3 and leveraging the Nvidia A100-PCIE-40GB as support, MAE, RMSE, and MAPE are used as metrics for comparison.

❷ **Datasets & Baselines.** We conduct a comprehensive evaluation of our model on six spatio-temporal datasets spanning multiple years across two domains. These datasets include LargeST (Liu et al., 2024b) and PEMSD3-Stream (Chen et al., 2021) in the traffic domain, and KnowAir (Wang et al., 2020) in the atmospheric domain. The dataset summary is presented in Table 1. Our comparison involves advanced **spatio-temporal prediction, spatio-temporal OOD learning, and spatio-temporal continual learning model**. ① Spatio-temporal predictionn models include STGCN (Yu et al., 2017), GWNet (Wu et al., 2019), STNorm (Deng et al., 2021), STID (Shao et al., 2022a), STAEformer (Liu et al., 2023a), STNN (Yang et al., 2021), D$^2$STGNN (Shao et al., 2022b), BigST (Han et al., 2024a), and RPMixer (Yeh et al., 2024). ② Spatio-temporal OOD learning models include CaST (Xia et al., 2024) and STONE (Wang et al., 2024a). ③ Spatio-temporal continual learning models include TrafficStream (Chen et al., 2021), PEMCP (Wang et al., 2023b), and TFMoE (Lee & Park, 2024). *Some models require the removal of non-essential components (such as node embedding in STID or adaptive graph learning method in GWNet) to adapt them to the ST-OOD setting, as the parameters of them are intertwined with the scale of the graph structure*, as elaborated in Appendix E.1.

❸ **ST-OOD Datasets.** For the evaluation of temporal shift, we train the models using data from the first year and test them on each subsequent year. The training set comprises the first 60% of data from the initial year dataset, while the following 20% of data is used as the validation set. In each subsequent year, the last 20% of data is designated as the test set. This setup aims to accentuate the temporal distribution difference between the test and training sets, while maintaining a ratio of approximately 6:2:2 for the training, validation, and test sets. Regarding structural shift evaluation, we select a subset of nodes for training and validation. In the test set, we decrease the number of nodes by 10% and introduce 30% new nodes to simulate shifts in the graph structure and scale. More detailed settings can be found in Appendix E.2.

## 4.2. OOD Performance Comparison(Q.1)

As shown in Table 3, we report the average values across all years of test sets on four datasets. Experiments on large

datasets can be found in Appendix E.3, and detailed year-specific reports can be found in Appendix E.9. The comparison with spatio-temporal continuous learning models is shown in Table 11.

GCN-based models like STGCN and GWNet underperform in OOD settings due to their reliance on the global messaging mechanism of GCN, rendering them highly sensitive to spatio-temporal shifts. Transformer-based models such as STAEformer and D$^2$STGNN exhibit improved predictive accuracy by leveraging self-attention mechanisms to aggregate global node features, effectively addressing spatio-temporal shift errors. Despite these advancements, STGNNs still face challenges in generalizing weights for unseen graph structures. On the other hand, spatio-temporal OOD learning baselines like STONE introduce diverse training environments utilizing perturbation-generated semantic relations to learn invariant causal knowledge, resulting in enhanced performance. STOP demonstrates significant improvements across various metrics, with a maximum enhancement of **17.01%**. This improvement can be attributed to its robust centralized messaging mechanism, which facilitates effective spatial feature interaction.

## 4.3. Node-to-node Interaction vs. Ours (Q.2)

We used two representative backbone models, STGCN and STAEformer, to compare the effectiveness of different message mechanisms. The former utilizes graph convolution for inter-node interaction, while the latter employs self-attention mechanisms for node-to-node interaction. We removed their node-to-node message mechanisms and labeled these variants as "-graph". Additionally, we replaced their inter-node interactions with our spatial interaction mechanism, denoting these variants as "+Ours". Using the SD and KnowAir datasets with OOD settings, the performance results are shown in Table 2. We can find that variants without node-to-node message mechanisms performed better than the original models, indicating that the initial mechanism limited the models' generalization performance. Furthermore, these models achieve generalization performance improvements after integrating our centralized messaging mechanism. Therefore, this further validates the effectiveness and necessity of our central interaction mechanism.

*Table 2.* Comparison of two interaction mechanisms.

| Variant | SD | | | KnowAir | | |
|---|---|---|---|---|---|---|
| | MAE | RMSE | MAPE | MAE | RMSE | MAPE |
| STGCN | 25.72 | 40.03 | 18.21 | 29.49 | 40.93 | 63.85 |
| STGCN - graph | 25.45 | 39.62 | 17.98 | 26.18 | 38.03 | 55.75 |
| STGCN + Ours | **24.87** | **38.98** | **17.65** | **25.44** | **37.42** | **52.80** |
| STAEformer | 26.20 | 41.18 | 18.39 | 27.25 | 38.93 | 56.48 |
| STAEformer - graph | 25.80 | 40.84 | 17.45 | 25.82 | 37.28 | 55.65 |
| STAEformer + Ours | **24.65** | **38.46** | **17.30** | **25.46** | **37.25** | **55.04** |

*Table 3.* OOD performance comparisons on four datasets. The unit of MAPE is percent (%). We bold the best-performing model results in **red** and underline the sub-optimal model results in blue.

| | | Method | **Ours** | STONE | CaST | RPMixer | BigST | D$^2$STGNN | STNN | STAEformer | STID | STNorm | GWNet | STGCN |
|---|---|---|---|---|---|---|---|---|---|---|---|---|---|---|
| SD | 3 | MAE | **17.71** | 18.44 | 21.35 | 24.92 | 18.56 | 18.70 | 36.46 | 18.70 | 19.68 | 18.82 | 20.15 | 18.68 |
| | | RMSE | **28.45** | 29.55 | 33.28 | 39.88 | 29.93 | 29.31 | 56.84 | 28.97 | 29.56 | 30.06 | 31.34 | 29.61 |
| | | MAPE | **11.73** | 12.32 | 16.04 | 15.63 | 12.18 | 13.04 | 26.91 | 12.62 | 13.18 | 12.82 | 14.44 | 12.92 |
| | 6 | MAE | **23.62** | 25.10 | 29.28 | 42.37 | 25.66 | 25.13 | 36.91 | 25.80 | 25.87 | 26.00 | 28.07 | 25.25 |
| | | RMSE | **37.71** | 39.66 | 45.24 | 66.45 | 40.61 | 38.77 | 57.59 | 40.73 | 40.86 | 41.20 | 43.00 | 39.48 |
| | | MAPE | **15.99** | 17.56 | 21.49 | 26.15 | 18.03 | 17.46 | 27.15 | 17.59 | 18.03 | 18.03 | 21.17 | 17.34 |
| | 12 | MAE | **32.59** | 37.12 | 42.40 | 77.31 | 37.89 | 36.35 | 41.69 | 37.17 | 38.30 | 38.08 | 39.75 | 36.15 |
| | | RMSE | **51.82** | 54.60 | 64.05 | 115.62 | 58.74 | 53.60 | 64.99 | 57.81 | 59.40 | 59.24 | 61.08 | 55.74 |
| | | MAPE | **22.89** | 25.90 | 31.73 | 49.48 | 27.12 | 25.98 | 31.32 | 27.07 | 26.90 | 27.89 | 31.46 | 26.41 |
| GBA | 3 | MAE | **18.33** | 20.19 | 21.85 | 24.79 | 19.92 | 19.10 | 40.61 | 20.91 | 19.09 | 20.86 | 20.65 | 21.49 |
| | | RMSE | **29.70** | 33.65 | 34.32 | 39.59 | 32.33 | 32.64 | 60.07 | 33.59 | 31.40 | 32.92 | 32.21 | 33.57 |
| | | MAPE | **13.64** | 15.10 | 18.61 | 17.06 | 14.75 | 14.29 | 33.77 | 14.93 | 14.36 | 16.00 | 15.70 | 14.79 |
| | 6 | MAE | **24.75** | 25.84 | 29.70 | 40.77 | 28.64 | 26.10 | 40.50 | 28.61 | 26.90 | 31.24 | 28.39 | 30.05 |
| | | RMSE | **38.48** | 41.96 | 45.16 | 62.24 | 43.93 | 41.72 | 59.96 | 44.03 | 42.15 | 46.69 | 42.60 | 44.97 |
| | | MAPE | **20.48** | 21.24 | 25.77 | 29.48 | 22.25 | 21.26 | 33.68 | 22.41 | 21.79 | 25.57 | 22.74 | 22.84 |
| | 12 | MAE | **34.93** | 39.56 | 42.60 | 72.51 | 42.87 | 36.26 | 44.62 | 41.68 | 39.36 | 45.73 | 39.61 | 43.29 |
| | | RMSE | **53.10** | 56.18 | 63.33 | 104.93 | 63.06 | 56.23 | 65.61 | 62.28 | 59.60 | 65.62 | 58.33 | 62.34 |
| | | MAPE | **31.09** | 32.18 | 36.88 | 56.28 | 34.52 | 32.23 | 38.28 | 34.99 | 33.43 | 41.02 | 33.67 | 35.23 |
| PEMSD3-Stream | 3 | MAE | **11.39** | 13.27 | 15.43 | 14.68 | 12.79 | 12.89 | 17.04 | 12.81 | 12.96 | 13.03 | 12.97 | 13.39 |
| | | RMSE | **19.48** | 21.48 | 24.53 | 23.73 | 20.79 | 21.14 | 28.47 | 21.02 | 20.95 | 21.07 | 21.11 | 21.60 |
| | | MAPE | **15.45** | 17.06 | 32.15 | 18.02 | 17.30 | 16.58 | 23.63 | 16.48 | 16.66 | 20.44 | 16.41 | 16.71 |
| | 6 | MAE | **12.47** | 14.30 | 17.13 | 17.41 | 14.05 | 14.08 | 17.26 | 14.14 | 14.18 | 14.51 | 14.14 | 14.63 |
| | | RMSE | **21.62** | 23.68 | 27.63 | 28.61 | 23.07 | 23.26 | 29.27 | 23.38 | 23.19 | 23.67 | 23.31 | 23.82 |
| | | MAPE | **16.02** | 18.23 | 33.77 | 20.90 | 19.54 | 17.62 | 25.63 | 19.71 | 18.52 | 22.43 | 17.91 | 18.33 |
| | 12 | MAE | **14.36** | 16.28 | 20.96 | 24.00 | 16.65 | 16.55 | 18.19 | 16.71 | 16.56 | 17.04 | 16.37 | 17.25 |
| | | RMSE | **24.95** | 28.41 | 33.82 | 39.64 | 27.46 | 27.44 | 30.14 | 27.92 | 27.31 | 27.94 | 27.10 | 28.20 |
| | | MAPE | **18.66** | 20.94 | 39.07 | 27.84 | 23.59 | 20.12 | 30.81 | 20.95 | 21.25 | 25.30 | 20.29 | 21.30 |
| KnowAir | 6 | MAE | **24.37** | 25.68 | 26.20 | 30.56 | 26.89 | 26.43 | 27.85 | 26.19 | 26.49 | 28.46 | 27.84 | 27.92 |
| | | RMSE | **36.56** | 37.59 | 38.42 | 45.34 | 39.16 | 37.91 | 39.07 | 37.82 | 38.90 | 41.47 | 40.25 | 39.47 |
| | | MAPE | **51.94** | 52.41 | 59.53 | 69.06 | 57.45 | 58.39 | 65.74 | 52.90 | 57.84 | 65.26 | 52.42 | 58.32 |
| | 12 | MAE | **27.03** | 28.96 | 29.49 | 38.45 | 29.77 | 30.06 | 30.48 | 29.45 | 30.85 | 30.86 | 31.11 | 31.63 |
| | | RMSE | **40.29** | 42.64 | 41.98 | 55.26 | 41.75 | 42.52 | 42.67 | 41.71 | 44.59 | 43.87 | 43.65 | 43.71 |
| | | MAPE | **54.45** | 71.99 | 70.15 | 87.60 | 68.39 | 67.10 | 71.05 | 61.64 | 68.44 | 71.83 | 61.51 | 69.83 |
| | 24 | MAE | **28.70** | 30.56 | 31.63 | 42.67 | 31.57 | 30.94 | 31.48 | 30.96 | 32.78 | 32.52 | 32.99 | 34.68 |
| | | RMSE | **42.39** | 45.48 | 45.21 | 61.30 | 44.52 | 46.21 | 44.72 | 43.48 | 46.67 | 44.80 | 44.14 | 47.19 |
| | | MAPE | **57.96** | 75.11 | 75.36 | 94.76 | 76.76 | 69.84 | 74.14 | 65.31 | 74.02 | 81.32 | 70.84 | 80.49 |

## 4.4. Hyperparameter Sensitivity Analysis (Q.3)

We analyze the sensitivity of the numer of ConAU and GenPU on the SD (upper) and KnowAir (lower) datasets on the Figure 3. Hyperparameter experiments on other datasets are provided in Appendix Section E.11. When the number of ConAU $K$ is set to 8 in SD dataset and 4 in KnowAir dataset. When $K$ exceeds this value, the model creates too many ConAU, making it unable to focus on extracting invariant contextual features, thus introducing noise. When $K$ is less than this value, too few perception units fail to learn sufficient invariant knowledge, leading to a decrease in the model's generalization performance. The number of GenPU $M$ is set to 3 in SD dataset and 4 in KnowAir dataset. A smaller $M$ may not provide sufficient training environment diversity, resulting in performance degradation. On the other hand, an excessive number of GenPU does not necessarily improve performance. Too large $M$ means that the generated environment is too complex, which in-

creases the learning difficulty of the model to extract causal knowledge.

## 4.5. Ablation Study (Q.4)

In this section, we evaluate the effectiveness of each component on SD and KnowAir datasets. "w/o decom" removes the time decomposition module, "w/o prompt" eliminates the spatio-temporal prompting method, "w/o $\mathbf{Y}_t$" uses only spatial prediction as the final prediction. "w/o LA" means that we use vanilla self-attention mechanism to replace the low-rank attention module.

As illustrated in Figure 4, the results show that each component of STOP helps to improve the OOD generalization. "w/o $\mathbf{Y}_t$" achieves poor prediction performance, which proves that the proposed collaborative component is effective for OOD. "w/o ConAU" removes ConAU and achieves high errors, proving that spatio-temporal interaction is also

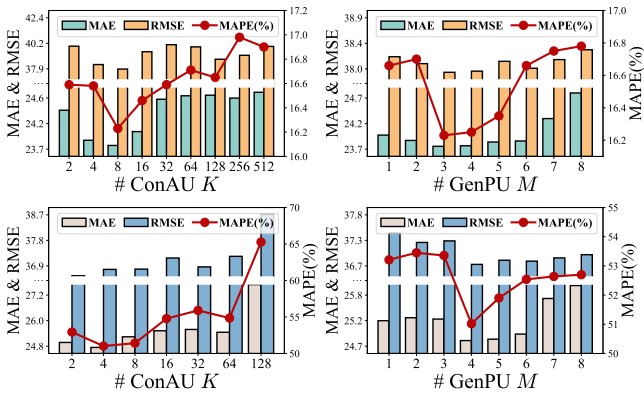

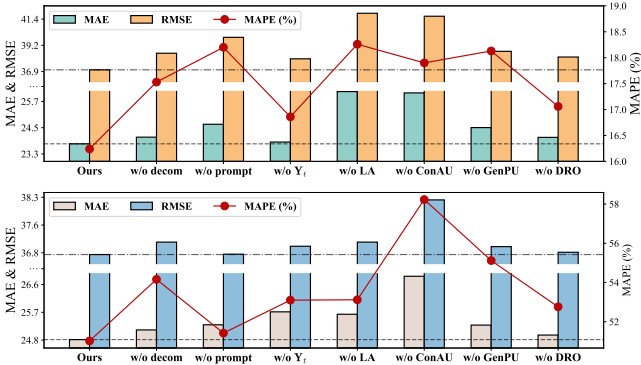

*Figure 3.* Hyperparameter Experiments for $K$ and $M$.

*Figure 4.* Ablation experiments on SD (upper) and KnowAir (lower) datasets.

necessary in OOD scenarios. "w/o GenPU" has higher prediction errors because GenPU can help the model extract causal knowledge and enhance model robustness. Ablation experiments demonstrate the necessity and effectiveness of each module of our model, while we further perform a comprehensive ablation experiment including double module ablation in Appendix Section E.10.

### 4.6. Efficiency Study (Q.5)

The training time of per epoch is illustrated in Figure 5, we can see that STOP demonstrates remarkable effectiveness and efficiency on the SD dataset. Since our model primarily uses lightweight MLP layers to model temporal and spatial dynamics. Compared to the SOTA model $D^2$STGNN, which is a Transformer-based model with RNNs for temporal dynamics modeling and GNNs for spatial dynamics modeling, our model has improved the efficiency by about 20 times (60.57 s/epoch vs 1220.79 s/epoch). Because its transformer-based architecture introduces a quadratic dependency on the number of spatial nodes, leading to high computational complexity. Furthermore, its recursive temporal modeling mechanism requires storing hidden representations at each time step, which significantly increases both memory consumption and inference latency. Consequently, such sequential approaches encounter substantial scalability barriers when applied to large-scale air quality forecasting tasks. In contrast, our model demonstrates superior computational efficiency. This is attributed to our novel spatiotemporal interaction module, which captures complex dynamics with near-linear time complexity. As a result, our approach enables highly scalable deployment.

### 4.7. Inductive Learning Performance of STOP (Q.6)

In OOD scenarios, the model's ability to represent unseen nodes is a challenge, which mainly involves the model's inductive learning. We report the performance of some ad-

vanced baselines for new nodes on SD and GBA datasets in Table 4 and on CA and GLA datasets in Table 7. Specifically, Transformer-based models, like $D^2$STGNN, are good at generalizing because they can create accurate representations for new nodes. GCN-based models are poor at generalizing because they rely heavily on the original graph structure and can not generate accurate representations for new nodes. Transformer-based models struggle because they can not create robust weights for new nodes. However, the STONE framework uses a novel embedding method to generate good initial representations for new nodes. Our model stands out by using a centralized messaging mechanism to access contextual features and enhance representations, making it excel in extending performance to new nodes.

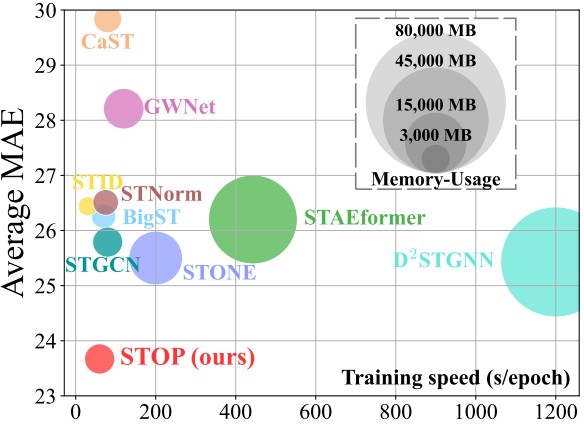

*Figure 5.* Efficiency study of STOP on SD dataset.

## 5. Related Work

### 5.1. Spatio-temporal Prediction

As a crucial task in intelligent transportation systems, current SOTA spatio-temporal prediction models primarily rely

*Table 4.* Inductive learning performance comparisons.

| Model | SD | | | GBA | | |
|---|---|---|---|---|---|---|
| | MAE | RMSE | MAPE | MAE | RMSE | MAPE |
| STONE | 25.06 | 39.15 | 18.12 | 26.28 | 42.02 | 24.12 |
| CaST | 28.83 | 44.10 | 22.93 | 29.62 | 44.97 | 25.56 |
| BigST | 25.22 | 39.22 | 18.02 | 28.75 | 43.80 | 24.32 |
| D²STGNN | 25.85 | 39.33 | 20.07 | 26.36 | 42.15 | 25.96 |
| STAEformer | 25.28 | 39.42 | 18.41 | 28.61 | 44.87 | 27.61 |
| STID | 25.40 | 39.53 | 18.66 | 26.83 | 43.94 | 24.42 |
| GWNet | 29.01 | 44.18 | 22.82 | 29.15 | 43.60 | 28.00 |
| **Ours** | **22.74** | **36.09** | **16.82** | **24.76** | **38.52** | **19.67** |

on STGNNs (Wang et al., 2024b;c; Zhang et al., 2025; 2024; Huang et al., 2023). However, they assume similarity between training and testing environments. To learn from evolving spatio-temporal graphs, some representative continual learning methods (Chen et al., 2021; Wang et al., 2023a) sequentially fine-tune models using data subsets with new distributions to adapt to spatio-temporal changes. Unfortunately, these models' effectiveness is only proven in environments similar to the training set, leading to challenges when encountering OOD scenarios. To overcome OOD challenges, researchers initially addressed temporal shift challenges, with models focusing on extracting invariant spatio-temporal knowledge from training data. However, they largely overlooked the evolution of spatio-temporal graph structures—which poses challenges for their relied-upon node-to-node interaction mechanisms. In this paper, we propose a novel message passing mechanism to overcome these challenges.

### 5.2. Spatio-temporal OOD Learning

Inspired by advances in time series shift learning (Liu et al., 2022) discussed in Appendix A, researchers have specifically designed spatio-temporal OOD learning models. For example, CauSTG (Zhou et al., 2023a) introduces a causal framework that transfers global invariant spatio-temporal relationships to OOD scenarios. CaST (Xia et al., 2023) employs a structural causal model to elucidate the data generation process of spatio-temporal graphs. STONE (Wang et al., 2024a) proposes a causal graph structure to learn robust spatio-temporal semantic relationship. STEVE (Hu et al., 2023) encodes traffic data into two disentangled representations and utilizes spatio-temporal environments as self-supervised signals. In this paper, we reformulate their message-passing mechanism, addressing the OOD challenge from a novel perspective.

### 5.3. Temporal Shift in Time Series

Various models have been developed in the time series domain to address temporal shifts in time series data, particularly focusing on OOD learning challenges. For in-

stance, RevIN (Kim et al., 2021) employs a symmetric structure to eliminate and reconstruct distribution information based on the input window's statistics. AdaRNN (Du et al., 2021) categorizes historical time sequences into different classes and dynamically matches input data to these classes for contextual information identification. Additionally, a reversible instance normalization technique, proposed by (Kim et al., 2021), aims to mitigate temporal distribution shift issues. Non-stationary Transformers (Liu et al., 2022) introduce a normalization-denormalization technique to stabilize time series data, mainly for transformer-based models. SAF (Arik et al., 2022) suggests test-time adaptation through a self-supervised objective to enhance adaptation against distribution shifts. DIVERSIFY (Lu et al., 2024) aims to leverage subdomains within a dataset to mitigate challenges arising from non-stationary generalized representation learning. However, these models often overlook the modeling of spatial dependencies. Spatial modeling is crucial in the field of spatio-temporal prediction, as it can examine the states of neighboring nodes to enhance prediction performance, given the strong correlations that often exist among neighboring nodes (Jin et al., 2023; Shao et al., 2023).

## 6. Conclusion

In this paper, we propose a Spatio-temporal Out-of-distribution Processor (STOP), which combines spatial interaction mechanisms and message perturbation mechanisms to enhance adaptation to spatio-temporal variations. We employ a centralized message passing mechanism and message shape perturbation mechanism to replace traditional point-to-point message interaction strategies. STOP can learn generalizable knowledge from diverse training environments. Evaluation on extensive datasets across various OOD scenarios demonstrates the model's robust generalization, inductive learning capabilities, and efficiency.

## Impact Statement

This paper presents research aimed at advancing the field of Machine Learning. While there are many potential societal consequences of our work, we believe that none require specific emphasis in this context.

## Acknowledgment

This paper is partially supported by the National Natural Science Foundation of China (No.12227901). The AI-driven experiments, simulations and model training were performed on the robotic AI-Scientist platform of Chinese Academy of Sciences.

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

# A. Related Work

## A.1. Spatio-temporal Prediction

Current popular spatio-temporal prediction models are predominantly based on spatio-temporal graph neural networks (STGNNs) (Zhang et al., 2016; Wang et al., 2024b; Zhou et al., 2023c; Liu et al., 2025a; Pan et al., 2019; Wang et al., 2023c; Zhou et al., 2023b; Ma et al., 2025). These models focus on developing advanced variants to accurately characterize spatio-temporal data, typically combining GCNs with sequential models to learn complex dynamics. GWNet (Wu et al., 2019), STGCN (Yu et al., 2017) leverage GNNs with TCN to model spatial and temporal dynamics, respectively. D$^2$STGNN (Shao et al., 2022b), DGCRN (Li et al., 2023) and DCRNN (Li et al., 2017) integrate diffusion graph convolutional networks with RNN to effectively capture temporal patterns. Meanwhile, STAEformer (Liu et al., 2023a) and STNN (Yang et al., 2021) utilize Transformer to model long-term temporal dependencies. AGCRN (Bai et al., 2020) directly treats the graph topology as a trainable parameter of the model and goes through the training process to find the optimal graph topology. In addition, some spatio-temporal models choose other ways of spatial interaction, such as STNorm (Deng et al., 2021) which adjusts the data distribution of all nodes by normalization to accomplish spatial information influence among nodes, and STID (Shao et al., 2022a) captures and distinguishes spatial patterns of each node in a data-driven manner by assigning learnable node embeddings. However, these kind of spatial modeling on nodes has no capability of out-of-distribution learning since these parameters in models are coupled with the spatial scale. Furthermore, some continual learning approaches (Chen et al., 2021) sequentially fine-tune models using data subsets with new distributions to adapt to spatio-temporal changes. Unfortunately, the effectiveness of these models can only be demonstrated in environments similar to the training set, leading to challenges when encountering OOD scenarios.

## A.2. Continual Learning with Spatio-temporal Shift

Several studies (Chen et al., 2021; Miao et al., 2024; Wang et al., 2023a; Miao et al., 2025; Chen & Liang, 2024) have proposed continual learning strategies to tackle spatio-temporal graph prediction in scenarios with spatio-temporal shifts. When the spatio-temporal data distribution undergoes changes, these models engage in fine-tuning using a subset of new data to adjust to the updated data distribution. For instance, TrafficStream (Chen et al., 2021) recommends utilizing subsets of newly added nodes and significant temporal pattern data changes for fine-tuning the model. PECPM (Wang et al., 2023b) identifies conflicting nodes to enhance the fine-tuning process, focusing on nodes that have experienced substantial changes. DLF (Wang et al., 2024b) introduces a streaming training strategy to continuously fine-tune the model to adapt to the dynamic nature of spatio-temporal data. TFMoE (Lee & Park, 2024) partitions traffic flow into multiple homogeneous groups and assigns an expert model responsible for each group, enabling each expert model to specialize in learning and adapting to specific patterns. However, these models often compromise performance to improve learning efficiency, resulting in lower performance compared to traditional spatio-temporal models. Primarily, these models train and fine-tune on a sufficient amount of new distribution data (approximately 21 days in one month) and test on the new data distribution, thereby adhering to the IID assumption and encountering difficulties in OOD learning.

# B. Algorithm & Optimization

We have provided the pseudocode of the algorithm in Algorithm 1, where we can observe that STOP makes final predictions based on the temporal component and spatial component. This includes a perturbation process to extract robust knowledge. This perturbation process only occurs in the training phase and we no longer use it in the test phase. We also provide the optimization flow of GenPU and model parameters in Algorithm 2. As shown, we interleaved the optimization of GenPU and model parameters.

# C. Theoretical Explanation and Analysis

## C.1. Low-rank Attention

In the centralized messaging mechanism, We first define low-rank attention as follows:

$$\mathbf{Z}_c^{(i)} = \mathcal{A}\left(\mathbf{Q}, \mathbf{K}, \mathbf{V}\right) = \underbrace{\mathrm{softmax}\left(\alpha \mathbf{Q}\mathbf{K}^\top\right)}_{\textbf{Diffusion}} \times \underbrace{\mathrm{softmax}\left(\alpha \mathbf{K}\mathbf{Q}^\top\right)}_{\textbf{Aggregation}} \mathbf{V}, \tag{21}$$

$$\text{where} \quad \mathbf{Q} = \mathbf{Z}_\mathrm{T}\mathbf{W}_q^{(i)} \in \mathbb{R}^{N \times d_h}, \ \mathbf{K} = \mathbf{C}\mathbf{J}_{d_t}^{(i)} \in \mathbb{R}^{N \times d_h}, \ \mathbf{V} = \mathbf{Z}_\mathrm{T}\mathbf{J}_{d_t}^{(i)} \in \mathbb{R}^{N \times d_h}. \tag{22}$$

---

**Algorithm 1** STOP for spatio-temporal prediction

---

**Input:** Historical data $\mathbf{X} \in \mathbb{R}^{T \times N \times c}$.
**Output:** Future prediction $\widehat{\mathbf{Y}} \in \mathbb{R}^{T_P \times N \times c}$.
**# Data encode;**
$\mathbf{H}_0 \leftarrow \mathbf{X}$ in Eq. 2 $\sim$ 4      $\triangleright$ Temporal decomposition
$\mathbf{Z}_\mathrm{I} \leftarrow \mathbf{H}_0, \widetilde{\mathbf{T}}, \mathbf{E}$ in Eq. 5      $\triangleright$ Input representation
**# Temporal modeling and prediction;**
$\mathbf{Z}_\mathrm{T} \leftarrow \mathbf{Z}_\mathrm{I}$ in Eq. 6      $\triangleright$ Temporal representation learning
$\mathbf{Y}_t \leftarrow \mathbf{Z}_\mathrm{T}$ in Eq. 7      $\triangleright$ Temporal prediction component
**# Spatial modeling and prediction;**
**if test phase then**
    $\mathbf{Z}_c \leftarrow \mathbf{Z}_\mathrm{T}, \mathbf{C}$ in Eq. 8 $\sim$ 9      $\triangleright$ ConAU
**end if**
**if training phase then**
    $\mathbf{Z}_c \leftarrow \mathbf{Z}_\mathrm{T}, \mathbf{C}, \boldsymbol{g}$ in Eq. 8 $\sim$ 9, 10 $\sim$ 11      $\triangleright$ ConAU & GenPU
    $\mathbf{Z}_s \leftarrow \mathbf{Z}_c, \mathbf{Z}_\mathrm{T}$ in Eq. 15 $\sim$ 18      $\triangleright$ Spatial representation learning
    $\mathbf{Y}_s \leftarrow \mathbf{Z}_s$ in Eq. 19      $\triangleright$ Spatial prediction component
    **# Final prediction;**
    $\widehat{\mathbf{Y}} \leftarrow \mathbf{Y}_t + \mathbf{Y}_s$ in Eq. 20      $\triangleright$ Final prediction
**end if**

---

**Algorithm 2** Optimization process of STOP

---

**Input:** Historical data $\mathbf{X} \in \mathbb{R}^{T \times N \times c}$, GenPU $\mathbb{G} = \{\boldsymbol{g}_1, \boldsymbol{g}_2, \ldots, \boldsymbol{g}_M\} \subseteq \mathbb{R}^N$, sample hits $s \in (0, N)$, future label $\mathbf{Y} \in \mathbb{R}^{T_P \times N \times c}$, loss function $\mathcal{L}$, initialized parameters $\Theta$ of STOP function $f$, patience $P$, learning rates $\alpha$ and $\beta$.
**Output:** Well-trained parameters $\Theta^*$ of STOP.
**while** maximum epochs not reached or not converged **do**
    **for** patience $= 1, 2, \ldots, P$ **do**
        **for** $j = 1, 2, \ldots, M$ **do**
            $\boldsymbol{g}'_j \leftarrow \mathrm{softmax}\left(\boldsymbol{g}_j\right)$
            $\widetilde{\boldsymbol{g}}_j \leftarrow$ sampling from multinomial distribution $\mathcal{M}\left(\boldsymbol{g}'_j; s\right)$
            $\mathbf{G}_j \leftarrow \widetilde{\boldsymbol{g}}_j$ in Eq. 10      $\triangleright$ Generalized Perturbation Units
            $\mathcal{L}_j \leftarrow \mathcal{L}\left(f\left(\mathbf{X}\right), \mathbf{Y}; \mathbf{G}_j\right)$
        **end for**
        $\mathcal{L}^* \leftarrow \max\{\mathcal{L}_1, \mathcal{L}_2, \ldots, \mathcal{L}_M\}$
        $\Theta \leftarrow \Theta - \alpha \nabla_\Theta \mathcal{L}^*$      $\triangleright$ Update the parameters of STOP
    **end for**
    $i \leftarrow \mathrm{argmax}\{\mathcal{L}_1, \mathcal{L}_2, \ldots, \mathcal{L}_M\}$
    $\boldsymbol{g}_i \leftarrow \boldsymbol{g}_i + \beta\left(\left(\mathbf{1} - \widetilde{\boldsymbol{g}}_i\right)\boldsymbol{g}_i^\top - \log\|\exp\boldsymbol{g}_i\|_1\right)\mathcal{L}^*$      $\triangleright$ Update GenPU
**end while**

---

Let $\mathbf{S}_a = \mathrm{softmax}\left(\alpha\mathbf{K}\mathbf{Q}^\top\right) \in \mathbb{R}^{K \times N}$ be the aggregation component of the attention score, and $\mathbf{S}_d = \mathrm{softmax}\left(\alpha\mathbf{Q}\mathbf{K}^\top\right) \in \mathbb{R}^{N \times K}$ be the diffusion component of the attention score, hence the attention score matrix $\mathbf{S} \in \mathbb{R}^{N \times N}$ can be expressed as

$$\mathbf{S} = \mathbf{S}_d \times \mathbf{S}_a \in \mathbb{R}^{N \times N}. \tag{23}$$

And the rank of $\mathbf{S}$ is satisfied,

$$\mathrm{rank}\left(\mathbf{S}\right) = \mathrm{rank}\left(\mathbf{S}_d \times \mathbf{S}_a\right) \leq \min\left(\mathrm{rank}\left(\mathbf{S}_d\right), \mathrm{rank}\left(\mathbf{S}_a\right)\right) \leq K \ll N. \tag{24}$$

The final inequality is a consequence of the fact that the maximum rank of a matrix is no more than the minimum of the ranks of its rows and columns (Greub, 2012). The rank of $\mathbf{S}$, up to $K$, is much lower than its size $N$, i.e., the number of rows and columns, hence the attention score matrix of our attention mechanism is a low-rank matrix. This constitutes the basis for the low ranking observed in our low-rank attention mechanism.

The low-rank characteristic in the centralized messaging mechanism offers one key advantage: it exhibits linear complexity compared to the self-attention mechanism, allowing for a larger spatio-temporal efficiency.

### C.1.1. EFFICIENCY ANALYSIS

The low-rank attention function in Equation 21 can be rewritten as follows,

$$\mathcal{A}\left(\mathbf{Q}, \mathbf{K}, \mathbf{V}\right) = \mathbf{S}\mathbf{V} = \left(\mathbf{S}_d\mathbf{S}_a\right) \times \mathbf{V} = \mathbf{S}_d \times \left(\mathbf{S}_a\mathbf{V}\right). \tag{25}$$

Consequently, in contrast to the unlike vanilla self-attention mechanism (Vaswani et al., 2017), which necessitates the pre-computation of the attention score matrix with complexity $\mathcal{O}\left(N^2 d_h\right)$, we have the option of computing $\mathbf{S}_a\mathbf{V} \in \mathbb{R}^{K \times d_h}$ initially with complexity $\mathcal{O}\left(KN d_h\right)$ and subsequently determining $\mathbf{S}_d \times \left(\mathbf{S}_a\mathbf{V}\right) \in \mathbb{R}^{N \times d_h}$ with same complexity $\mathcal{O}\left(KN d_h\right)$, resulting in the efficient computation of low-rank attention with linear time complexity $\mathcal{O}\left(N\right)$ by $K \ll N$. As shown, we reduce the computational complexity from quadratic to nearly linear. This enables our method to effectively process graph data with a large number of nodes without requiring excessive GenPU memory resources. See Figure 9 and Figure 6 for experimental analysis.

## D. Distributionally Robust Optimization

We theoretically analyzed STOP's generalization performance. Since STOP's optimization objective belongs to the distributionally robust optimization class (Duchi & Namkoong, 2019), which exhibits good generalization properties. Note that distributionally robust optimization class is a general term for optimization objectives that satisfy specific conditions - our contribution lies in how to implement optimization strategies that meet these conditions in the spatio-temporal OOD problem. First, we will introduce what constitutes a distributionally robust optimization class and the necessary conditions for membership, then analyze its beneficial properties, and finally extend these concepts to STOP.

### D.1. What is DRO?

Distributionally Robust Optimization (DRO) (Duchi & Namkoong, 2019) refers to a class of loss functions that aim to optimize by considering the worst-case scenario within a certain range of all possible distributions of the data. In practical terms, an optimization object that takes the following form with respect to the training environment $e^*$ can be categorized under DRO (Duchi & Namkoong, 2019; Staib & Jegelka, 2019; Levy et al., 2020),

$$\operatorname{argmin}_f \sup_{e \in \mathcal{E}} \left\{\mathbb{E}_{(\mathbf{X}, \mathbf{Y}) \sim p(\mathcal{X}, \mathcal{Y}|e); \mathcal{D}(e, e^*) \leq \rho}\left[\mathcal{L}\left(f\left(\mathbf{X}\right), \mathbf{Y}\right)\right]\right\}, \tag{26}$$

where $f$ is the function we optimized, usually a deep neural network with learnable parameters. $\mathcal{D}\left(\cdot, \cdot\right)$ is the distribution distance metric (Namkoong & Duchi, 2016; Shafieezadeh Abadeh et al., 2018), which is used to calculate the distance between distributions. $\rho$ is a hyperparamer to limit the extent to which the distribution is explored. In fact, this merely indicates an exploration constraint, limiting the explored environments from being completely different from the training environment.

**Mark**. If an optimization satisfies: (1) modeling of different environments, (2) applying constraints, and (3) emphasizing the most challenging environments, then this optimization belongs to DRO and possesses the following beneficial properties.

### D.2. Robust Properties of DRO

Recall that in the preliminary, the task of spatio-temporal OOD learning aims to learn a robust function $f$, which can accurately predict values after $T_P$ time steps given observed data of past $T$ time steps $\mathbf{X}$ and the graph sampled from any environment $e \sim \mathcal{E}$, where $e$ may have different spatio-temporal distributions with training environment $e^*$,

$$\operatorname{argmin}_f \sup_{e \in \mathcal{E}} \mathbb{E}_{(\mathbf{X}, \mathbf{Y}) \sim p(\mathcal{X}, \mathcal{Y}|e)}\left[\mathcal{L}\left(f\left(\mathbf{X}\right), \mathbf{Y}\right)\right]. \tag{27}$$

In a more intuitive sense, Equation. 1 is designed to find a function that reduces the loss associated with the most challenging scenario across all possible distributions $e \sim \mathcal{E}$. This task is particularly challenging because we lack access to data from any unfamiliar distributions outside of the training set (Qiao & Peng, 2023). Although traditional Empirical Risk Minimisation (Vapnik, 1998),

$$\operatorname{argmin}_f \mathbb{E}_{(\mathbf{X}, \mathbf{Y}) \sim p(\mathcal{X}, \mathcal{Y}|e^*)}\left[\mathcal{L}\left(f\left(\mathbf{X}\right), \mathbf{Y}\right)\right], \tag{28}$$

which optimises solely based on the raw training environment $e^*$, performs well under the IID assumption, it is not possible to guarantee its performance in the presence of distributional drifts (Arjovsky et al., 2019). For all possible $e \in \mathcal{E}$ and function $f$, with high probability in mathematics, the following property holds,

$$\mathbb{E}_{(\mathbf{X},\mathbf{Y}) \sim p(\mathcal{X},\mathcal{Y}|e); \mathcal{D}(e,e^*) \leq \rho} \left\{ [\mathcal{L}\left(f\left(\mathbf{X}\right), \mathbf{Y}\right)] \right\}$$
$$\leq \mathbb{E}_{(\mathbf{X},\mathbf{Y}) \sim p(\mathcal{X},\mathcal{Y}|e^*)} \left[ \mathcal{L}\left(f\left(\mathbf{X}\right), \mathbf{Y}\right) \right] + \mathcal{O}\left( \sqrt{\frac{\mathrm{Var}_{(\mathbf{X},\mathbf{Y}) \sim p(\mathcal{X},\mathcal{Y}|e^*)} \left[ \mathcal{L}\left(f\left(\mathbf{X}\right), \mathbf{Y}\right) \right]}{N_{e^*}}} \right), \tag{29}$$

where $N_{e^*}$ is the number of data point in traning environment. Therefore, due to the presence of subsequent variance terms, optimizing ERM alone cannot guarantee performance improvement in other environments $e' \in \mathcal{E} - \{e^*\}$. Compared to the IID-only condition of the ERM, distributionally robust optimization explores a certain range of challenging training data distributions, mathematically, distributionally robust optimization is equivalent to adding variance regularization to the standard ERM (Duchi & Namkoong, 2019),

$$\sup_{e \in \mathcal{E}} \left\{ \mathbb{E}_{(\mathbf{X},\mathbf{Y}) \sim p(\mathcal{X},\mathcal{Y}|e); \mathcal{D}(e,e^*) \leq \rho} \left[ \mathcal{L}\left(f\left(\mathbf{X}\right), \mathbf{Y}\right) \right] \right\}$$
$$= \mathbb{E}_{(\mathbf{X},\mathbf{Y}) \sim p(\mathcal{X},\mathcal{Y}|e^*)} \left[ \mathcal{L}\left(f\left(\mathbf{X}\right), \mathbf{Y}\right) \right] + \sqrt{2\rho \, \mathrm{Var}_{(\mathbf{X},\mathbf{Y}) \sim p(\mathcal{X},\mathcal{Y}|e^*)} \left[ \mathcal{L}\left(f\left(\mathbf{X}\right), \mathbf{Y}\right) \right]} + \varepsilon\left(f\right), \tag{30}$$

where $\varepsilon\left(f\right) \geq 0$ and it is $\mathcal{O}(1/N_{e^*})$ uniformly about $f$. Therefore, if we do not consider the subsequent asymptotic terms $\epsilon(f)$, the above formula is equivalent to the following inequality,

$$\sup_{e \in \mathcal{E}} \left\{ \mathbb{E}_{(\mathbf{X},\mathbf{Y}) \sim p(\mathcal{X},\mathcal{Y}|e); \mathcal{D}(e,e^*) \leq \rho} \left[ \mathcal{L}\left(f\left(\mathbf{X}\right), \mathbf{Y}\right) \right] \right\}$$
$$\geq \mathbb{E}_{(\mathbf{X},\mathbf{Y}) \sim p(\mathcal{X},\mathcal{Y}|e^*)} \left[ \mathcal{L}\left(f\left(\mathbf{X}\right), \mathbf{Y}\right) \right] + \sqrt{2\rho \, \mathrm{Var}_{(\mathbf{X},\mathbf{Y}) \sim p(\mathcal{X},\mathcal{Y}|e^*)} \left[ \mathcal{L}\left(f\left(\mathbf{X}\right), \mathbf{Y}\right) \right]}. \tag{31}$$

DRO explores a certain range of training data distributions and tries to optimise on data distributions that may match the distribution of the test set, providing ideas for solving the OOD problem. Therefore, DRO mathematically provides more rigorous constraints than using empirical loss functions alone in OOD environments, preventing the model from over-relying on training data. This enables the model to flexibly adapt to different environments, improving its generalization performance in unknown environments.

### D.3. Does STOP have properties of DRO?

We will demonstrate that our optimization objective of STOP belongs to DRO, inheriting its good properties. Our optimization objective is as follows:

$$\min_{f} \sup_{\boldsymbol{g} \in \mathbb{R}^N} \mathbb{E}_{(\mathbf{X},\mathbf{Y}) \sim (\mathcal{X},\mathcal{Y}|e^*)} \left[ \mathcal{L}\left(f\left(\mathbf{X}\right), \mathbf{Y}; \boldsymbol{g}\right) \right], \quad \text{s.t. } ||\widetilde{\boldsymbol{g}}||_0 = s \in (0, N). \tag{32}$$

Next, we demonstrate according to Mark 1 that our proposed optimization strategy satisfies the necessary conditions for DRO, thus inheriting its beneficial properties.

**Diverse environments**. STOP creates a diverse training environment by adding a perturbation process through a message perturbation mechanism.

**Applying constraints**. Our perturbation process follows polynomial distribution sampling, and we strictly control the perturbation ratio, which imposes constraints on the generated environments.

**Exploring challenging environments:** We emphasize selecting environments with the largest gradients during training for optimization, encouraging the model to be exposed to challenging environments.

In summary, our optimization strategy belongs to DRO and thus inherits its good generalization property.

## E. Experiments

### E.1. Baseline Details

In experiments, we compare a lot of spatio-temporal prediction models with spatio-temporal OOD models. However, the original versions of many of these models are not compatible with the OOD setting. Consequently, we had to remove certain

non-essential code related to graph structures, particularly node embedding techniques and adaptive graph structure learning techniques.

❶ **Node embedding technology.** The researchers set a node embedding vector $E \in \mathbb{R}^{N \times d_s}$ to capture node patterns adaptively, which are coupled with the size $N$ of the graph structure. Therefore, when the model is trained, it cannot be run directly into the test environment with ST-OOD. STID, STAEformer, and BigST use this technology.

❷ **Adaptive graph learning.** This method generally use two noode embedding vectors $E_1 \in \mathbb{R}^{N \times d_s}$ and $E_2 \in \mathbb{R}^{N \times d_s}$, and they multiply these two node embedding matrices, $A_s = E_1 E_2^\top \in \mathbb{R}^{N \times N}$, to generate an adaptive adjacency matrix $A_s$ for learning the adjacency matrix, which is then used for GCN. GWNet, D²STGNN, and CaST adopt this method.

## E.2. Experimental dataset details

In this paper, we utilized six datasets to evaluate the effectiveness of the models in OOD scenarios, primarily from the domains of transportation and atmosphere. These datasets often span multiple years. Among them, LargeST (Liu et al., 2024b) collected five years of data from 8600 records, sampled at a frequency of five minutes. PEMSD3-Stream (Chen et al., 2021) is a naturally streaming traffic dataset, recording data from July each year from 2011 to 2017, where the traffic structure expands year by year, naturally representing spatio-temporal shifts. Knowair (Wang et al., 2020) collected 18 atmospheric features sampled at an hourly frequency. We followed the following rules to create spatio-temporal OOD datasets.

❶ **Temporal shift**: We used the first 60% of data from the first year as the training set, followed by 20% of data for the validation set. We used the last 20% of data from subsequent years for the test set. This longer time interval ensures changes in temporal distribution characteristics.

❷ **Structural shift**: Apart from the PEMSD3-Stream dataset, we selected a subset of nodes for training and validation, approximately 75% of the total number, in the test set, we randomly masked 10% of nodes to simulate node disappearance and added 30% of nodes as new nodes. This is because for spatio-temporal systems, cities or detection systems generally tend to expand. Since PEMSD3-Stream is a natural streaming data set, we use it directly.

*Table 5.* The details of used datasets.

| Dataset | Training set | | Test set | | |
| --- | --- | --- | --- | --- | --- |
| | Time range | Graph Nodes | Temporal shift | Structural shift | |
| | | | | New nodes | Removed Nodes |
| LargeST-SD | First 60% data in 2017 | 550 | Last 20% data in 2018-2021 | 165 | 55 |
| LargeST-GBA | First 60% data in 2017 | 1809 | Last 20% data in 2018-2021 | 542 | 180 |
| LargeST-GLA | First 60% data in 2017 | 2949 | Last 20% data in 2018-2021 | 884 | 294 |
| LargeST-CA | First 60% data in 2017 | 6615 | Last 20% data in 2018-2021 | 1984 | 661 |
| KnowAir | First 60% data in 2011 | 141 | Last 20% data in 2012-2017 | 42 | 14 |
| PEMSD3-Stream | First 60% data in 2015 | 655 | Last 20% data in 2016-2021 | (60, 131, 167, 179, 195, 216) | 0 |

## E.3. OOD Performance Comparison on Large Datasets

As the largest collection of spatio-temporal data available in open source today, CA represents an invaluable test case for the OOD capability of the model. The performance of STOP and the baseline is evaluated on large-scale and large-scale spatio-temporal datasets, respectively, under identical conditions.

Based on the same partitioning strategy as described in Section 4.1, we divide the LargeST dataset into the two largest subdatasets, GLA and CA. Due to the parameter complexity of Transformer-based baselines such as STAEformer, STNN, D²STGNN, and STONE, which scales at least quadratically with the number of nodes, deploying these models on GLA and CA datasets is not feasible.

As shown in Table 6, STOP consistently outperforms the baselines on both the large-scale spatio-temporal OOD dataset in terms of overall performance and performance on newly added nodes, with improvements of up to 14.01%. On large-scale spatio-temporal datasets, the performance of baselines based on global message passing mechanisms declines significantly due to the introduction of more new nodes. STID, which does not involve node interactions, achieves the second-best

performance among the baselines. In contrast, STOP benefits from ConAU by decomposing large-scale spatio-temporal scenes into stable spatio-temporal subenvironments, leading to the best performance while ensuring node interactions. This highlights STOP's remarkable OOD capabilities even in large-scale scenarios.

*Table 6.* OOD performance comparisons on GLA and CA datasets. The absence of baselines indicates that the models incur out-of-memory issues.

| | | Method | **Ours** | CaST | RPMixer | BigST | STID | STNorm | GWNet | STGCN |
|---|---|---|---|---|---|---|---|---|---|---|
| GLA | 3 | MAE | **19.13** | 23.36 | 25.89 | 20.32 | 19.87 | 21.05 | 21.17 | 20.51 |
| | | RMSE | **30.33** | 35.53 | 41.10 | 32.56 | 31.78 | 33.03 | 32.96 | 32.24 |
| | | MAPE | **11.93** | 21.44 | 14.90 | 12.93 | 12.03 | 13.34 | 13.87 | 12.81 |
| | 6 | MAE | **26.29** | 31.43 | 43.33 | 28.83 | 28.30 | 30.70 | 29.91 | 29.13 |
| | | RMSE | **40.66** | 47.49 | 66.65 | 44.69 | 43.92 | 46.35 | 45.47 | 44.50 |
| | | MAPE | **17.60** | 27.75 | 26.18 | 18.49 | 17.72 | 20.57 | 19.90 | 19.36 |
| | 12 | MAE | **36.87** | 43.48 | 77.32 | 42.12 | 41.38 | 46.13 | 41.81 | 43.92 |
| | | RMSE | **55.96** | 65.08 | 114.02 | 62.99 | 62.69 | 66.98 | 62.08 | 64.34 |
| | | MAPE | **27.07** | 36.46 | 53.23 | 30.33 | 27.90 | 34.63 | 28.21 | 31.14 |
| CA | 3 | MAE | **17.47** | 21.87 | 23.72 | 18.77 | 18.35 | 19.10 | 19.01 | 19.23 |
| | | RMSE | **28.24** | 34.44 | 38.43 | 30.77 | 30.01 | 30.86 | 30.30 | 30.89 |
| | | MAPE | **12.69** | 17.79 | 16.02 | 13.60 | 12.92 | 15.38 | 13.62 | 13.68 |
| | 6 | MAE | **23.70** | 29.13 | 39.52 | 26.80 | 26.06 | 27.63 | 26.64 | 27.30 |
| | | RMSE | **37.17** | 45.30 | 61.88 | 42.34 | 41.33 | 43.10 | 41.32 | 42.51 |
| | | MAPE | **18.39** | 23.63 | 27.42 | 19.98 | 19.33 | 23.24 | 19.56 | 20.23 |
| | 12 | MAE | **32.86** | 41.26 | 70.64 | 39.59 | 38.23 | 40.77 | 37.63 | 40.64 |
| | | RMSE | **50.28** | 62.85 | 105.36 | 60.24 | 59.16 | 61.20 | 57.07 | 61.01 |
| | | MAPE | **27.65** | 34.71 | 53.26 | 32.00 | 29.77 | 35.50 | 30.31 | 31.93 |

*Table 7.* Inductive learning preformance on GLA and CA datasets of new nodes. The absence of baselines indicates that the models incur out-of-memory issues.

| | | Method | **Ours** | CaST | RPMixer | BigST | STID | STNorm | GWNet | STGCN |
|---|---|---|---|---|---|---|---|---|---|---|
| GLA | 3 | MAE | **18.99** | 23.09 | 25.65 | 20.17 | 19.71 | 20.92 | 21.35 | 20.36 |
| | | RMSE | **30.13** | 35.16 | 40.89 | 32.36 | 31.55 | 32.84 | 33.30 | 32.05 |
| | | MAPE | **11.94** | 21.32 | 14.86 | 12.91 | 12.03 | 13.26 | 14.01 | 12.81 |
| | 6 | MAE | **26.17** | 31.15 | 42.95 | 28.66 | 28.14 | 30.56 | 30.46 | 29.00 |
| | | RMSE | **40.57** | 47.16 | 66.35 | 44.51 | 43.76 | 46.24 | 46.51 | 44.39 |
| | | MAPE | **17.64** | 27.62 | 26.09 | 18.47 | 17.73 | 20.47 | 20.25 | 19.38 |
| | 12 | MAE | **36.78** | 43.12 | 76.60 | 41.84 | 41.13 | 45.87 | 42.97 | 43.70 |
| | | RMSE | **55.89** | 64.65 | 113.41 | 62.56 | 62.36 | 66.70 | 63.92 | 64.04 |
| | | MAPE | **27.17** | 36.35 | 53.12 | 30.24 | 27.87 | 34.35 | 28.98 | 31.14 |
| CA | 3 | MAE | **17.48** | 21.86 | 23.73 | 18.76 | 18.35 | 19.10 | 19.38 | 19.23 |
| | | RMSE | **28.39** | 34.50 | 38.59 | 30.86 | 30.14 | 30.98 | 30.87 | 30.96 |
| | | MAPE | **12.87** | 18.46 | 16.15 | 13.85 | 13.12 | 16.06 | 15.62 | 13.97 |
| | 6 | MAE | **23.71** | 29.11 | 39.50 | 26.79 | 26.05 | 27.65 | 27.47 | 27.30 |
| | | RMSE | **37.29** | 45.31 | 62.03 | 42.38 | 41.40 | 43.20 | 42.50 | 42.53 |
| | | MAPE | **18.73** | 24.37 | 27.65 | 20.34 | 19.70 | 24.45 | 22.76 | 20.62 |
| | 12 | MAE | **32.83** | 41.22 | 70.53 | 39.53 | 38.18 | 40.75 | 39.27 | 40.61 |
| | | RMSE | **50.30** | 62.78 | 105.38 | 60.14 | 59.13 | 61.20 | 59.43 | 60.94 |
| | | MAPE | **28.24** | 35.75 | 53.61 | 32.69 | 30.44 | 37.24 | 35.64 | 32.53 |

## E.4. Inductive Learning Comparison on Large Datasets

To evaluate the inductive learning capabilities of each model, we further report the performance of added nodes in Table 7. We can see that GCN-based models have overall poor inductive capabilities. While they can rely on message passing

mechanisms to generalize learned information to unseen nodes, the spatially confused interactions cannot guarantee accurate descriptions of added nodes, leading to subpar performance. In this regard, STID achieves better predictive results because it assumes nodes are independent, allowing the model to learn time-related knowledge that is unrelated to nodes, which can generalize to added nodes and avoid error accumulation. Our model demonstrates strong inductive learning capabilities on large-scale graphs, as added nodes can access shared context features to obtain good representations.

### E.5. Performance in S-OOD and T-OOD Scenarios

With LargeST-SD dataset, we investigate the performance of models in T-OOD and S-OOD scenarios. Used two datasets are simplified versions of ST-OOD. For S-OOD, we use the last 20% of the 2017 data as the test set with the graph structure unchanged. For T-OOD, we maintain the graph structure consistent between the training and testing environments, aligning the data selection with ST-OOD. The experimental results are shown in Table 8, and we can observe that STGNNs exhibit poor performance in the S-OOD scenario, mainly due to the sensitivity of the node-to-node interaction method to structural shifts. The poor performance of STNN can be attributed to its use of Transformer, which lacks robustness against noise introduced by temporal and spatial shifts. Our model has achieved competitive performance in both T-OOD and S-OOD scenarios.

*Table 8.* Comparison in T-OOD and S-OOD scenarios.

| | Method | **Ours** | STONE | D$^2$STGNN | STNN | STGCN | GWNet |
|---|---|---|---|---|---|---|---|
| | MAE | **23.21** | 25.00 | 26.56 | 35.06 | 29.74 | 26.79 |
| S-OOD | RMSE | **36.95** | 39.12 | 42.77 | 55.12 | 44.45 | 41.47 |
| | MAPE | **14.45** | 16.72 | 19.80 | 23.42 | 21.79 | 18.16 |
| | MAE | **22.91** | 25.41 | 24.23 | 36.14 | 25.73 | 23.38 |
| T-OOD | RMSE | **37.17** | 37.56 | 39.04 | 56.26 | 40.07 | 37.63 |
| | MAPE | **15.35** | 16.38 | 17.37 | 26.46 | 17.68 | 16.58 |

### E.6. Performance on Rapid Evoluting Spatio-temporal Dynamical System

In the main experiment, the proportion of added nodes is relatively small (only 30%), which may not cover rapidly developing urban scenarios. We further create a challenging scenario where we train on 30% of nodes from the year 2017 and test on the remaining 70% of nodes from subsequent years. Details of the experimental dataset are provided in Table 9.

*Table 9.* Rapidly growth OOD setting on SD dataset.

| Training set | | Test set | |
|---|---|---|---|
| Time range | Graph (Nodes) | Temporal shift | Strucal shift |
| Firtst 60% data in 2017 | 214 | Last 20% data in 2018-2021 | 500 new nodes & 0 removed nodes |

We observe that for baseline models based on Transformer and GCN, such as D$^2$STGNN and GWNet, the rapid and large influx of new nodes significantly disrupts the model's learning of message passing mechanisms, leading to a decrease in performance for models relying on such global message passing mechanisms. Models like BigST based on linear attention mechanisms and STONE based on relaxed mapping perform better than the former in out-of-distribution (OOD) scenarios with rapid growth. On the other hand, STID, based on node independence, shows limitations in generalizing features to new nodes when faced with a large number of additional nodes. In contrast, STOP benefits from its innovative ConAU and GenPU-oriented low-order attention mechanism, capturing flexible adaptations to changes in the overall spatio-temporal environment through sub-environments, showing the highest relative improvement rate at 16.35% and demonstrating robustness in scenarios with rapid node growth.

### E.7. Compare Continuous Learning Method

We compared STOP with several continual learning methods on out-of-distribution (OOD) tasks. Taking the PEMSD3-Stream dataset as an illustration, when encountering spatio-temporal shifts, these models require fine-tuning using 21-day data from the new distribution. To ensure a fair comparison, we aligned the OOD task settings by conducting tests directly in the subsequent years following the initial year of training. This training methodology is denoted as 'static-STModel' in

*Table 10.* OOD performance with rapidly growth on SD dataset.

| | Method | | **Ours** | STONE | CaST | RPMixer | BigST | D²STGNN | STNN | STAEformer | STID | STNorm | GWNet | STGCN |
|---|---|---|---|---|---|---|---|---|---|---|---|---|---|---|
| All nodes | 3 | MAE | **18.04** | 18.61 | 21.47 | 25.20 | 18.85 | 20.98 | 42.24 | 18.99 | 18.78 | 19.14 | 22.62 | 20.61 |
| | | RMSE | **29.17** | 29.82 | 33.75 | 40.13 | 30.36 | 33.46 | 65.12 | 30.29 | 30.17 | 30.84 | 34.70 | 32.56 |
| | | MAPE | **12.32** | 13.74 | 15.92 | 15.64 | 12.59 | 14.50 | 33.68 | 14.87 | 12.91 | 15.79 | 17.83 | 15.15 |
| | 6 | MAE | **23.64** | 25.03 | 28.80 | 42.69 | 26.32 | 30.83 | 42.67 | 26.68 | 26.52 | 26.60 | 32.67 | 28.28 |
| | | RMSE | **38.10** | 39.92 | 44.71 | 66.85 | 41.79 | 47.76 | 65.66 | 41.82 | 41.93 | 42.20 | 49.31 | 44.40 |
| | | MAPE | **16.49** | 18.89 | 20.73 | 26.13 | 17.79 | 21.61 | 34.18 | 22.79 | 18.89 | 22.20 | 25.61 | 20.54 |
| | 12 | MAE | **32.29** | 38.97 | 41.95 | 77.90 | 38.60 | 45.12 | 46.48 | 38.76 | 39.44 | 38.59 | 48.05 | 40.62 |
| | | RMSE | **51.74** | 55.20 | 63.40 | 116.56 | 60.11 | 68.19 | 71.04 | 59.68 | 60.79 | 60.35 | 71.78 | 63.28 |
| | | MAPE | **22.95** | 26.80 | 31.80 | 49.35 | 26.72 | 31.83 | 36.88 | 33.16 | 30.24 | 35.07 | 40.56 | 29.35 |
| New nodes | 3 | MAE | **18.28** | 18.84 | 21.53 | 25.24 | 18.97 | 21.26 | 44.82 | 19.16 | 18.92 | 19.37 | 22.97 | 20.86 |
| | | RMSE | **29.47** | 30.11 | 33.67 | 39.93 | 30.34 | 33.81 | 68.42 | 30.43 | 30.22 | 31.06 | 35.27 | 32.98 |
| | | MAPE | **12.50** | 14.07 | 16.20 | 15.66 | 12.72 | 14.98 | 35.29 | 15.21 | 13.08 | 16.38 | 18.45 | 15.54 |
| | 6 | MAE | **24.02** | 25.39 | 28.94 | 42.76 | 26.55 | 31.34 | 45.26 | 27.00 | 26.79 | 27.02 | 33.27 | 28.71 |
| | | RMSE | **38.57** | 40.39 | 44.74 | 66.81 | 41.94 | 48.51 | 68.97 | 42.19 | 42.17 | 42.74 | 50.15 | 45.06 |
| | | MAPE | **16.77** | 19.40 | 21.04 | 26.13 | 18.02 | 22.34 | 35.86 | 23.34 | 19.22 | 23.22 | 26.58 | 21.08 |
| | 12 | MAE | **32.81** | 39.40 | 42.19 | 77.95 | 38.86 | 45.73 | 48.99 | 39.22 | 39.78 | 39.24 | 48.98 | 41.26 |
| | | RMSE | **52.31** | 55.68 | 63.55 | 116.44 | 60.25 | 68.94 | 74.23 | 60.24 | 61.08 | 61.32 | 72.91 | 63.97 |
| | | MAPE | **23.33** | 27.39 | 32.22 | 49.28 | 26.88 | 32.76 | 38.59 | 33.87 | 30.86 | 37.24 | 41.97 | 30.12 |

TrafficStream (Chen et al., 2021), 'SurSTG-Static' in PEMCP (Wang et al., 2023b), and 'Static-TFMoE' in TFMoE (Lee & Park, 2024). We directly extracted their experimental results from the PEMSD3-Stream dataset. For an intuitive comparison, we have added the predicted performance of STGCN.

As depicted in Table 11, the performance of continual learning strategies is notably inferior to traditional prediction models because they trade performance for accelerated training processes. And our model significantly surpasses existing continual learning models in OOD tasks.

It is noteworthy that in this experiment, the performance indicated by STOP is slightly superior to that in the primary experiment because *the results amalgamate the performance of testing data in the first year*, which was omitted in the primary experiment to emphasize the disparities in data distribution between the test and training sets as much as possible.

*Table 11.* Compared with spatio-temporal continuous learning methods on PEMSD3-Stream dataset.

| Model | 15min | | | 30min | | | 60min | | |
|---|---|---|---|---|---|---|---|---|---|
| | MAE | RMSE | MAPE | MAE | RMSE | MAPE | MAE | RMSE | MAPE |
| PECMP | 13.37 | 21.10 | 28.35 | 14.78 | 23.54 | 30.88 | 16.32 | 27.20 | 34.28 |
| TrafficStream | 13.98 | 21.88 | 29.36 | 15.12 | 23.98 | 31.67 | 17.46 | 28.01 | 36.44 |
| TFMoE | 12.95 | 21.18 | 18.97 | 14.51 | 23.90 | 19.62 | 18.07 | 29.87 | 24.92 |
| STGCN | 13.27 | 21.03 | 16.64 | 14.47 | 23.64 | 18.03 | 17.05 | 27.95 | 21.04 |
| Ours | **11.37** | **19.16** | **15.38** | **12.41** | **21.18** | **15.92** | **14.24** | **24.39** | **18.51** |

## E.8. Efficiency Study on KnowAir Dataset

The training time per epoch is depicted in Figure 6, showcasing the remarkable effectiveness and efficiency of STOP on the KnowAir dataset. Transformer-based models like STNN, STARformer, and D²STGNN exhibit substantial computational time and high memory usage due to their utilization of self-attention mechanisms to calculate dependencies between node pairs, resulting in a time and space complexity that scales quadratically with the number of nodes. Similarly, GCN-based models rely on GCN mechanisms for spatial feature interactions, leading to a time complexity that is also quadratic with the number of nodes. In contrast, our model, with a complexity linear with the number of nodes, significantly reduces the computational complexity.

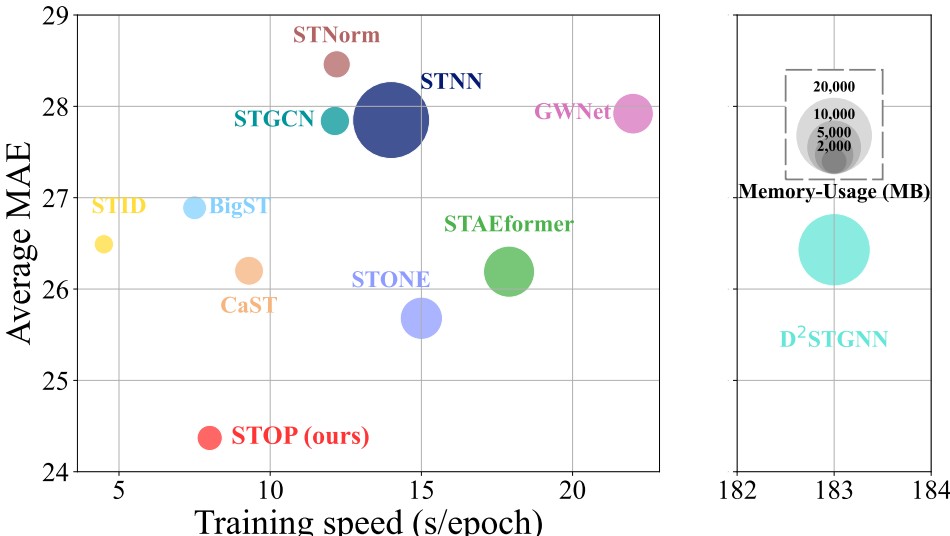

*Figure 6.* Visual case and efficiency study of STOP on KnowAir dataset.

### E.9. Detailed Performance Analysis of OOD in Each Year

In the main experiment, we reported the average OOD performance over multiple years. To provide a more detailed comparison, we present the performance changes of each model in LargeST OOD datasets for each year. As shown in Table 12 to 14, the results demonstrate that in fine-grained performance analysis, our model remains highly effective.

### E.10. Ablation Experiment

We conduct thorough ablation experiments to evaluate the effectiveness of each component. The variants we created are shown in Table 15 and the experiments are shown in Table 16.

For the time module, we found that time decomposition and prompting provided the model with better capabilities to capture the temporal patterns from the sequence perspective, while the introduction of $\mathbf{Y}_t$ to make predictions from multiple components enhanced the model's robustness.

Regarding the C&S messaging mechanism, the "w/o ConAU" variant, which removes the spatial interaction module, resulted in a significant increase in error, indicating that the spatial interaction is still necessary in OOD scenarios. The "w/o LA" variant, which removes the low-rank attention mechanism in the C&S spatial interaction module, performed poorly in prediction, as the traditional node-to-node messaging mechanism is less robust to spatio-temporal shifts. The "w/o LA + DRO" variant performed better than the "w/o LA + RandomDrop" variant, demonstrating that the proposed message perturbation mechanism is more effective than directly perturbing the dataset to generate diverse training environments in helping the model extract robust representations.

The "w/o DRO" variant exhibited a larger prediction error, suggesting that the inability to effectively optimize the deployed GenPU mask matrix increased the complexity of the model's learning process. The "w/o (GenPU&DRO)" variant also showed a considerable increase in error, further highlighting the crucial importance of the proposed message perturbation mechanism in enhancing the model's robustness, as it allows the model to learn resilient representations from the perturbed environments.

These ablation studies can demonstrate the positive impact of each designed component on enhancing the overall performance of the model in out-of-distribution scenarios.

*Table 12.* OOD performance on LargeST-2018 dataset

| | | Method | **Ours** | CaST | RPMixer | BigST | STID | STNorm | GWNet | STGCN | STONE | D²STGNN | STNN | STAEformer |
|---|---|---|---|---|---|---|---|---|---|---|---|---|---|---|
| SD | 3 | MAE | **17.80** | 22.07 | 26.22 | 19.13 | 18.65 | 19.48 | 19.84 | 18.68 | 18.83 | 18.38 | 35.32 | 19.01 |
| | | RMSE | **28.23** | 34.38 | 41.83 | 30.76 | 29.87 | 30.75 | 30.62 | 29.29 | 29.97 | 28.51 | 55.12 | 30.25 |
| | | MAPE | **10.76** | 14.68 | 15.28 | 11.47 | 12.25 | 11.96 | 12.50 | 11.56 | 11.24 | 11.47 | 23.38 | 11.58 |
| | 6 | MAE | **23.40** | 30.03 | 44.55 | 26.41 | 26.01 | 26.76 | 26.97 | 24.53 | 26.38 | 23.98 | 35.77 | 26.21 |
| | | RMSE | **37.13** | 46.41 | 69.75 | 41.64 | 40.92 | 41.64 | 41.36 | 38.01 | 38.53 | 37.72 | 55.73 | 41.04 |
| | | MAPE | **14.52** | 19.82 | 25.81 | 16.28 | 16.57 | 16.71 | 17.98 | 15.22 | 15.02 | 15.03 | 23.72 | 16.22 |
| | 12 | MAE | **32.06** | 43.42 | 80.17 | 38.86 | 38.31 | 38.93 | 37.66 | 34.78 | 38.77 | 32.97 | 41.51 | 38.25 |
| | | RMSE | 51.45 | 66.43 | 120.41 | 60.05 | 59.78 | 60.02 | 58.90 | 53.80 | 51.94 | **51.27** | 64.44 | 59.22 |
| | | MAPE | **20.59** | 29.10 | 48.67 | 24.68 | 24.39 | 25.31 | 26.42 | 22.96 | 22.15 | 21.70 | 28.11 | 24.79 |
| GBA | 3 | MAE | **19.87** | 24.51 | 28.16 | 22.44 | 21.70 | 22.33 | 21.96 | 22.61 | 20.58 | 20.48 | 41.67 | 23.27 |
| | | RMSE | **32.71** | 38.63 | 45.26 | 36.86 | 35.55 | 36.07 | 34.62 | 36.12 | 36.49 | 35.06 | 62.49 | 37.74 |
| | | MAPE | **15.74** | 23.37 | 21.21 | 18.03 | 17.34 | 18.08 | 18.71 | 17.26 | 16.65 | 16.67 | 39.87 | 18.17 |
| | 6 | MAE | **25.44** | 32.60 | 45.23 | 30.73 | 29.52 | 30.96 | 28.76 | 30.21 | 26.15 | 26.05 | 41.57 | 31.18 |
| | | RMSE | **40.59** | 49.60 | 69.42 | 47.93 | 46.23 | 47.50 | 43.56 | 46.13 | 42.86 | 42.49 | 62.41 | 48.28 |
| | | MAPE | **23.19** | 32.35 | 37.04 | 27.13 | 27.55 | 27.63 | 26.81 | 26.59 | 24.30 | 24.63 | 39.73 | 27.59 |
| | 12 | MAE | **33.94** | 45.49 | 77.43 | 43.71 | 41.78 | 43.38 | 38.42 | 41.67 | 36.96 | 34.94 | 46.13 | 44.53 |
| | | RMSE | **53.45** | 67.92 | 113.45 | 65.75 | 63.68 | 64.22 | 57.12 | 61.36 | 56.02 | 54.96 | 68.65 | 66.98 |
| | | MAPE | **33.85** | 45.54 | 69.67 | 41.11 | 42.03 | 43.41 | 38.80 | 40.22 | 37.77 | 37.31 | 46.05 | 43.32 |
| GLA | 3 | MAE | **19.70** | 24.78 | 28.12 | 21.81 | 21.03 | 22.27 | 21.52 | 21.73 | | | | |
| | | RMSE | **31.31** | 37.70 | 44.37 | 34.75 | 33.54 | 34.76 | 33.47 | 33.99 | | | | |
| | | MAPE | **11.25** | 19.65 | 14.84 | 12.42 | 11.58 | 12.62 | 12.69 | 11.91 | | | | |
| | 6 | MAE | **26.38** | 33.17 | 47.11 | 30.71 | 29.39 | 31.64 | 29.57 | 30.22 | | | | |
| | | RMSE | **41.01** | 50.28 | 72.19 | 47.55 | 45.86 | 47.81 | 45.05 | 46.24 | | | | |
| | | MAPE | **16.20** | 25.56 | 26.16 | 17.46 | 16.73 | 18.70 | 17.76 | 17.50 | | | | |
| | 12 | MAE | **36.15** | 45.37 | 83.11 | 44.34 | 42.28 | 45.84 | 40.51 | 44.17 | | Out of Memory | | |
| | | RMSE | **55.61** | 68.36 | 121.95 | 66.37 | 64.77 | 67.03 | 60.64 | 65.32 | | | | |
| | | MAPE | **24.33** | 33.70 | 53.19 | 27.87 | 25.94 | 29.77 | 24.75 | 26.93 | | | | |
| CA | 6 | MAE | **18.66** | 23.86 | 26.15 | 20.59 | 20.07 | 20.41 | 19.96 | 20.84 | | | | |
| | | RMSE | **30.26** | 37.58 | 42.44 | 33.80 | 32.72 | 33.06 | 31.84 | 33.53 | | | | |
| | | MAPE | **13.24** | 18.53 | 17.31 | 14.28 | 13.71 | 14.50 | 14.02 | 14.24 | | | | |
| | 12 | MAE | **24.67** | 31.61 | 43.27 | 28.95 | 28.08 | 28.61 | 27.00 | 29.04 | | | | |
| | | RMSE | **39.12** | 49.24 | 67.85 | 45.85 | 44.60 | 45.05 | 41.99 | 45.49 | | | | |
| | | MAPE | **18.98** | 24.83 | 29.84 | 20.89 | 20.56 | 21.59 | 19.74 | 20.89 | | | | |
| | 24 | MAE | **33.54** | 44.03 | 75.81 | 41.51 | 40.52 | 40.85 | 37.02 | 42.04 | | | | |
| | | RMSE | **52.42** | 67.48 | 113.33 | 63.71 | 63.08 | 62.43 | 56.46 | 63.80 | | | | |
| | | MAPE | **27.78** | 36.25 | 57.70 | 32.88 | 31.63 | 32.32 | 30.10 | 32.09 | | | | |

*Table 13.* OOD performance on LargeST-2019 dataset

| | | Method | **Ours** | CaST | RPMixer | BigST | STID | STNorm | GWNet | STGCN | STONE | D²STGNN | STNN | STAEformer |
|---|---|---|---|---|---|---|---|---|---|---|---|---|---|---|
| SD | 3 | MAE | **18.42** | 22.51 | 26.61 | 19.71 | 19.51 | 19.56 | 21.12 | 19.63 | 19.42 | 19.54 | 37.94 | 19.52 |
| | | RMSE | **29.68** | 35.36 | 42.80 | 32.18 | 31.39 | 31.25 | 33.00 | 31.15 | 30.88 | 30.68 | 59.76 | 31.62 |
| | | MAPE | **11.73** | 16.02 | 16.03 | 12.27 | 13.18 | 12.65 | 14.57 | 12.99 | 12.76 | 12.91 | 28.17 | 12.58 |
| | 6 | MAE | **24.16** | 30.58 | 44.89 | 26.85 | 26.88 | 26.32 | 28.93 | 26.05 | 25.86 | 25.76 | 38.28 | 26.58 |
| | | RMSE | **38.73** | 47.64 | 70.91 | 43.07 | 42.76 | 41.85 | 44.61 | 40.84 | 40.54 | 39.74 | 60.37 | 42.38 |
| | | MAPE | **15.81** | 21.39 | 26.91 | 17.42 | 17.95 | 17.44 | 20.96 | 17.25 | 17.80 | 17.12 | 28.40 | 17.50 |
| | 12 | MAE | **32.78** | 43.96 | 80.12 | 39.17 | 39.31 | 38.24 | 40.49 | 37.07 | 36.89 | **35.59** | 43.02 | 38.22 |
| | | RMSE | **52.84** | 67.24 | 120.77 | 61.26 | 61.64 | 60.13 | 62.82 | 57.94 | 54.16 | 54.24 | 67.88 | 60.11 |
| | | MAPE | **22.22** | 31.30 | 50.17 | 26.46 | 26.42 | 26.47 | 30.52 | 26.08 | 26.00 | 25.06 | 32.33 | 26.42 |
| GBA | 3 | MAE | **19.95** | 23.90 | 27.07 | 21.52 | 21.01 | 21.98 | 21.95 | 22.43 | 22.26 | 21.46 | 41.80 | 22.59 |
| | | RMSE | **32.18** | 37.26 | 42.81 | 34.69 | 34.16 | 34.68 | 33.94 | 35.05 | 34.94 | 35.45 | 62.20 | 35.99 |
| | | MAPE | **15.45** | 21.66 | 19.51 | 16.54 | 15.95 | 17.48 | 17.87 | 16.16 | 16.04 | 16.84 | 38.21 | 16.74 |
| | 6 | MAE | **26.30** | 31.96 | 43.87 | 29.93 | 29.05 | 31.42 | 29.34 | 30.51 | 28.25 | 27.84 | 41.72 | 30.51 |
| | | RMSE | **40.91** | 48.34 | 66.45 | 45.96 | 45.13 | 46.99 | 43.72 | 45.62 | 42.30 | 43.83 | 62.10 | 46.78 |
| | | MAPE | **23.51** | 30.05 | 34.24 | 24.85 | 25.50 | 27.53 | 25.63 | 25.25 | 25.05 | 24.59 | 38.23 | 25.25 |
| | 12 | MAE | **36.07** | 44.88 | 76.42 | 43.52 | 41.75 | 44.80 | 39.71 | 43.12 | 36.93 | 37.50 | 45.94 | 43.56 |
| | | RMSE | **54.99** | 67.05 | 110.42 | 65.05 | 63.30 | 64.97 | 58.44 | 62.39 | 56.18 | 57.40 | 67.93 | 65.74 |
| | | MAPE | **34.75** | 42.42 | 65.49 | 37.91 | 38.94 | 43.22 | 37.05 | 38.59 | 35.69 | 36.41 | 43.50 | 38.83 |
| GLA | 3 | MAE | **19.69** | 24.47 | 27.31 | 21.23 | 20.76 | 21.51 | 21.69 | 21.10 | | | | |
| | | RMSE | **30.93** | 37.04 | 43.19 | 33.92 | 32.95 | 33.70 | 33.58 | 33.12 | | | | |
| | | MAPE | **11.74** | 21.06 | 15.06 | 12.69 | 12.01 | 12.88 | 13.66 | 12.35 | | | | |
| | 6 | MAE | **26.68** | 32.83 | 45.88 | 29.83 | 29.44 | 30.68 | 30.29 | 29.50 | | | | |
| | | RMSE | **41.06** | 49.58 | 70.56 | 46.45 | 45.54 | 46.59 | 45.96 | 45.29 | | | | |
| | | MAPE | **17.13** | 27.37 | 26.73 | 17.97 | 17.68 | 19.31 | 19.35 | 18.42 | | | | |
| | 12 | MAE | **36.79** | 45.09 | 81.30 | 42.90 | 42.73 | 44.89 | 41.81 | 43.33 | | | | |
| | | RMSE | **56.03** | 67.71 | 119.66 | 64.97 | 64.92 | 66.28 | 62.28 | 64.54 | | | | |
| | | MAPE | **25.92** | 36.07 | 55.21 | 28.69 | 27.67 | 31.03 | 27.11 | 28.64 | | Out of Memory | | |
| CA | 6 | MAE | **18.50** | 23.33 | 25.21 | 19.90 | 19.51 | 19.91 | 19.81 | 20.19 | | | | |
| | | RMSE | **29.65** | 36.55 | 40.72 | 32.59 | 31.74 | 32.19 | 31.40 | 32.40 | | | | |
| | | MAPE | **13.34** | 18.47 | 16.80 | 14.01 | 13.44 | 15.19 | 14.04 | 13.94 | | | | |
| | 12 | MAE | **24.94** | 30.93 | 41.86 | 28.08 | 27.58 | 28.22 | 27.27 | 28.23 | | | | |
| | | RMSE | **38.97** | 48.06 | 65.53 | 44.55 | 43.67 | 44.30 | 42.14 | 44.22 | | | | |
| | | MAPE | **19.55** | 24.73 | 29.04 | 20.55 | 20.30 | 22.95 | 20.04 | 20.56 | | | | |
| | 24 | MAE | **34.31** | 43.25 | 73.86 | 40.68 | 40.08 | 40.80 | 37.76 | 41.13 | | | | |
| | | RMSE | **52.57** | 66.25 | 110.23 | 62.63 | 62.31 | 62.15 | 57.27 | 62.72 | | | | |
| | | MAPE | **29.07** | 36.29 | 56.63 | 32.69 | 31.37 | 34.17 | 30.78 | 31.72 | | | | |

*Table 14.* OOD performance on LargeST-2021 dataset

| | | Method | **Ours** | CaST | RPMixer | BigST | STID | STNorm | GWNet | STGCN | STONE | D²STGNN | STNN | STAEformer |
|---|---|---|---|---|---|---|---|---|---|---|---|---|---|---|
| SD | 3 | MAE | **18.24** | 21.42 | 25.11 | 19.24 | 18.92 | 19.29 | 20.97 | 19.33 | 18.61 | 19.54 | 38.32 | 19.69 |
| | | RMSE | **29.23** | 33.16 | 39.71 | 30.70 | 30.10 | 30.74 | 32.61 | 30.60 | 30.01 | 30.73 | 58.72 | 27.31 |
| | | MAPE | **12.02** | 15.99 | 15.64 | 12.27 | 13.33 | 12.80 | 15.29 | 13.21 | 12.95 | 13.81 | 27.23 | 12.91 |
| | 6 | MAE | **24.22** | 29.31 | 42.46 | 26.47 | 26.50 | 26.78 | 29.11 | 26.33 | 26.09 | 26.55 | 38.64 | 27.21 |
| | | RMSE | **38.64** | 44.84 | 65.77 | 41.67 | 41.67 | 42.10 | 44.43 | 41.13 | 40.94 | 41.07 | 59.25 | 42.59 |
| | | MAPE | **16.31** | 21.48 | 26.12 | 18.57 | 18.41 | 17.96 | 21.94 | 17.82 | 17.58 | 18.35 | 27.34 | 18.10 |
| | 12 | MAE | **33.06** | 42.01 | 77.24 | 38.48 | 38.91 | 38.60 | 40.88 | 37.16 | 37.77 | 37.26 | 42.59 | 38.66 |
| | | RMSE | **52.31** | 62.75 | 114.49 | 59.15 | 60.06 | 59.64 | 62.34 | 57.35 | 56.07 | 56.52 | 65.51 | 59.78 |
| | | MAPE | **22.98** | 31.50 | 49.02 | 26.74 | 27.21 | 27.48 | 31.56 | 26.67 | 26.02 | 26.77 | 31.06 | 27.48 |
| GBA | 3 | MAE | **17.44** | 20.37 | 23.25 | 18.68 | 17.64 | 20.12 | 19.76 | 21.12 | 18.23 | 18.32 | 39.34 | 19.63 |
| | | RMSE | **28.33** | 32.32 | 37.24 | 30.49 | 29.58 | 31.67 | 30.91 | 32.83 | 29.56 | 31.67 | 57.83 | 31.84 |
| | | MAPE | 11.46 | 13.84 | 13.74 | 11.88 | 10.73 | 13.67 | 12.54 | 12.59 | **10.72** | 12.06 | 26.84 | 12.03 |
| | 6 | MAE | **24.12** | 27.81 | 38.87 | 27.59 | 25.21 | 31.23 | 27.70 | 30.22 | 25.87 | 25.42 | 39.18 | 26.96 |
| | | RMSE | **37.13** | 42.60 | 59.14 | 42.14 | 39.93 | 46.19 | 41.54 | 44.89 | 39.06 | 41.14 | 57.64 | 41.73 |
| | | MAPE | **17.16** | 19.18 | 23.40 | 17.63 | 16.38 | 22.01 | 18.15 | 18.99 | 17.21 | 17.75 | 26.75 | 17.39 |
| | 12 | MAE | **35.00** | 39.89 | 70.24 | 41.91 | 37.03 | 45.97 | 39.12 | 43.87 | 36.03 | 36.09 | 42.72 | 39.04 |
| | | RMSE | **52.48** | 59.20 | 100.88 | 60.59 | 56.12 | 64.95 | 57.59 | 62.31 | 58.97 | 55.86 | 62.52 | 58.17 |
| | | MAPE | **26.56** | 27.65 | 44.32 | 27.42 | 26.84 | 34.75 | 26.88 | 29.04 | 26.67 | 26.96 | 29.43 | 26.71 |
| GLA | 3 | MAE | **18.86** | 22.84 | 24.75 | 19.69 | 20.31 | 20.59 | 20.98 | 19.98 | Out of Memory | | | |
| | | RMSE | **30.05** | 34.85 | 39.70 | 31.81 | 32.13 | 32.59 | 32.77 | 31.76 | | | | |
| | | MAPE | **11.99** | 21.13 | 14.56 | 12.68 | 12.89 | 13.15 | 13.95 | 12.59 | | | | |
| | 6 | MAE | **26.22** | 30.69 | 41.37 | 27.88 | 28.59 | 30.16 | 29.73 | 28.42 | | | | |
| | | RMSE | **40.56** | 46.41 | 63.92 | 43.35 | 43.85 | 45.63 | 45.22 | 43.60 | | | | |
| | | MAPE | **18.04** | 27.41 | 25.56 | 18.23 | 18.60 | 20.36 | 20.05 | 19.14 | | | | |
| | 12 | MAE | **37.19** | 42.11 | 74.06 | 40.33 | 41.17 | 45.21 | 41.30 | 43.05 | | | | |
| | | RMSE | **56.17** | 62.84 | 110.23 | 60.29 | 61.56 | 65.44 | 61.33 | 62.86 | | | | |
| | | MAPE | **28.09** | 35.91 | 51.89 | 29.45 | 28.61 | 34.08 | 27.84 | 30.83 | | | | |
| CA | 6 | MAE | **16.89** | 21.15 | 22.68 | 17.93 | 17.58 | 18.46 | 18.47 | 18.46 | Out of Memory | | | |
| | | RMSE | **27.45** | 33.42 | 36.75 | 29.54 | 28.99 | 30.00 | 29.52 | 29.84 | | | | |
| | | MAPE | **11.87** | 16.48 | 14.87 | 12.56 | 12.02 | 15.25 | 12.76 | 12.60 | | | | |
| | 12 | MAE | **23.08** | 28.08 | 37.95 | 25.66 | 25.06 | 26.82 | 26.22 | 26.20 | | | | |
| | | RMSE | **36.07** | 43.74 | 59.24 | 40.57 | 39.82 | 41.83 | 40.67 | 40.85 | | | | |
| | | MAPE | **17.21** | 21.83 | 25.41 | 18.40 | 17.88 | 23.02 | 18.39 | 18.58 | | | | |
| | 24 | MAE | **32.27** | 39.63 | 68.35 | 37.99 | 36.71 | 39.46 | 37.29 | 39.01 | | | | |
| | | RMSE | **49.11** | 60.16 | 101.84 | 57.47 | 56.62 | 58.96 | 56.56 | 58.40 | | | | |
| | | MAPE | **26.04** | 31.94 | 49.28 | 29.39 | 27.37 | 34.64 | 28.40 | 29.37 | | | | |

*Table 15.* Variants and their definitions in ablation experiment.

| Variant | Definition |
|---|---|
| w/o decom | Remove the decoupling mechanism |
| w/o prompt | Remove the temporal prompt learning |
| w/o (decom & prompt) | Remove the decoupling mechanism and temporal prompt learning |
| w/o $\mathbf{Y}_t$ | Remove the temporal prediction component |
| w/o $\mathbf{Y}_s$ | Remove the spatio-temporal prediction component |
| w/o ConAU | Completely remove the spatial centralized messaging mechanism |
| w/o LA | Use naive self-attention mechanism to replace Low-rank attention |
| w/o LA + GenPU | Add GenPU term with the variant w/o LA |
| w/o LA + GenPU +DRO | Add GenPU and spatio-temporal DRO with the variant w/o LA |
| w/o LA + RandomDrop | Randomly mask 20% training nodes and then train variant w/o LA |
| w/o DRO | Remove spatio-temporal DRO |
| w/o (GenPU) | Remove spatio-temporal DRO and GenPU |
| w/o (GenPU&DRO) + RandomDrop | Remove spatio-temporal DRO and GenPU and randomly mask 20% training nodes to simulate temporal and spatial shifts |

*Table 16.* Ablation experiments on SD and KnowAir datasets.

| Variant | SD | | | KnowAir | | |
|---|---|---|---|---|---|---|
| | MAE | RMSE | MAPE | MAE | RMSE | MAPE |
| **Ours** | **23.79** | **37.94** | **16.24** | **24.78** | **36.77** | **51.02** |
| w/o decom | 24.09 | 38.49 | 17.53 | 25.10 | 37.10 | 54.16 |
| w/o prompt | 24.67 | 39.83 | 18.20 | 25.27 | 36.78 | 51.42 |
| w/o decom & prompt | 25.23 | 40.46 | 19.01 | 25.83 | 37.25 | 54.33 |
| w/o $\mathbf{Y}_t$ | 23.87 | 38.02 | 16.86 | 25.70 | 36.99 | 53.10 |
| w/o $\mathbf{Y}_s$ | 26.25 | 41.25 | 18.76 | 27.04 | 39.21 | 63.68 |
| w/o ConAU | 26.06 | 41.47 | 17.56 | 26.88 | 38.22 | 58.23 |
| w/o LA | 26.14 | 41.86 | 18.26 | 25.62 | 37.10 | 53.12 |
| w/o LA + GenPU | 26.29 | 42.15 | 18.71 | 25.61 | 36.86 | 55.81 |
| w/o LA + GenPU + DRO | 26.11 | 41.73 | 17.58 | 25.10 | 36.91 | 54.73 |
| w/o LA + RandomDrop | 27.41 | 43.11 | 18.32 | 25.77 | 37.16 | 59.09 |
| w/o DRO | 24.08 | 38.17 | 17.06 | 24.93 | 37.24 | 54.86 |
| w/o (GenPU&DRO) | 24.52 | 38.65 | 18.13 | 25.26 | 36.98 | 55.12 |
| w/o (GenPU&DRO) + RandomDrop | 24.77 | 38.90 | 18.48 | 25.45 | 36.87 | 55.90 |

### E.11. Hyperparameter Sensitivity Experiments

In addition to the hyperparameter experiment in Section 4.4 of the main body, we additionally deployed conduct experiments on four datasets—SD, GBA, GLA, and CA—to analyze the sensitivity of two hyperparameters, the number of ConAU $K$ and the number of GenPU $M$. The numebr of nodes *for training* in these six datasets range from 141 to 6615 nodes. The results on six datasets are shown in Figure 7 and Figure 8.

❶ **The number of ConAU** $K$. ConAU is the coarsening unit set up to interact with the node. Thus, the number of ConAU $K$ is closely related to the spatial scale. Based on our observations, we find that setting $K$ to approximately 1% of the spatial scale is a good choice. A larger number of ConAU can hinder the model's ability to focus on capturing generalizable contextual features.

❷ **The number of GenPU** $M$. The hyperparameter $M$ represents the number of GenPU, which are used to modulate the interaction process between nodes and ConAU. Each GenPU corresponds to a different training environment. We have observed that the number of GenPU $M$ is universally effective when set to between 2 and 4. When $M$ is set to a smaller value, an overly complex training environment can disrupt learning stability. Conversely, if there are too few GenPU, the limited training environments may not provide sufficient diversity for the model to extract invariant knowledge. Interestingly, this hyperparameter is insensitive to spatial scale.

We further analyze the sensitivity of this hyperparameter to the temporal span of the dataset. Long-range SD, GBA, GLA, and CA datasets contain a full year of training data, and TrafficStream is a short-range dataset containing one month data for training. And we can see that $M$ is not highly correlated with the time span of the data.

**Summary**. Based on the above analysis, we recommend setting the initial values $K$ to 1% of the number of training nodes and the initial values of $M$ between 2 and 4 for hyperparameter tuning in out-of-distribution (OOD) scenarios.

### E.12. Effectiveness Evaluation of Collaborative Structure

STOP's predictions arise from the superposition of spatial and temporal components, which helps improve the model's robustness across various OOD scenarios. Using the SD dataset as an example, we employ two separate scenarios: S-OOD and T-OOD. We use Shapley Values to analyze the contribution levels of $\mathbf{Y}_t$ and $\mathbf{Y}_s$ components in each scenario. The normalized results are shown in the following table: We observe that when structural shifts occur (S-OOD), the contribution of the temporal prediction component increases, due to the decreased accuracy of spatial correlation features. The opposite occurs in the T-OOD scenario. Through this collaborative architecture, our STOP can flexibly adapt to various OOD scenarios.

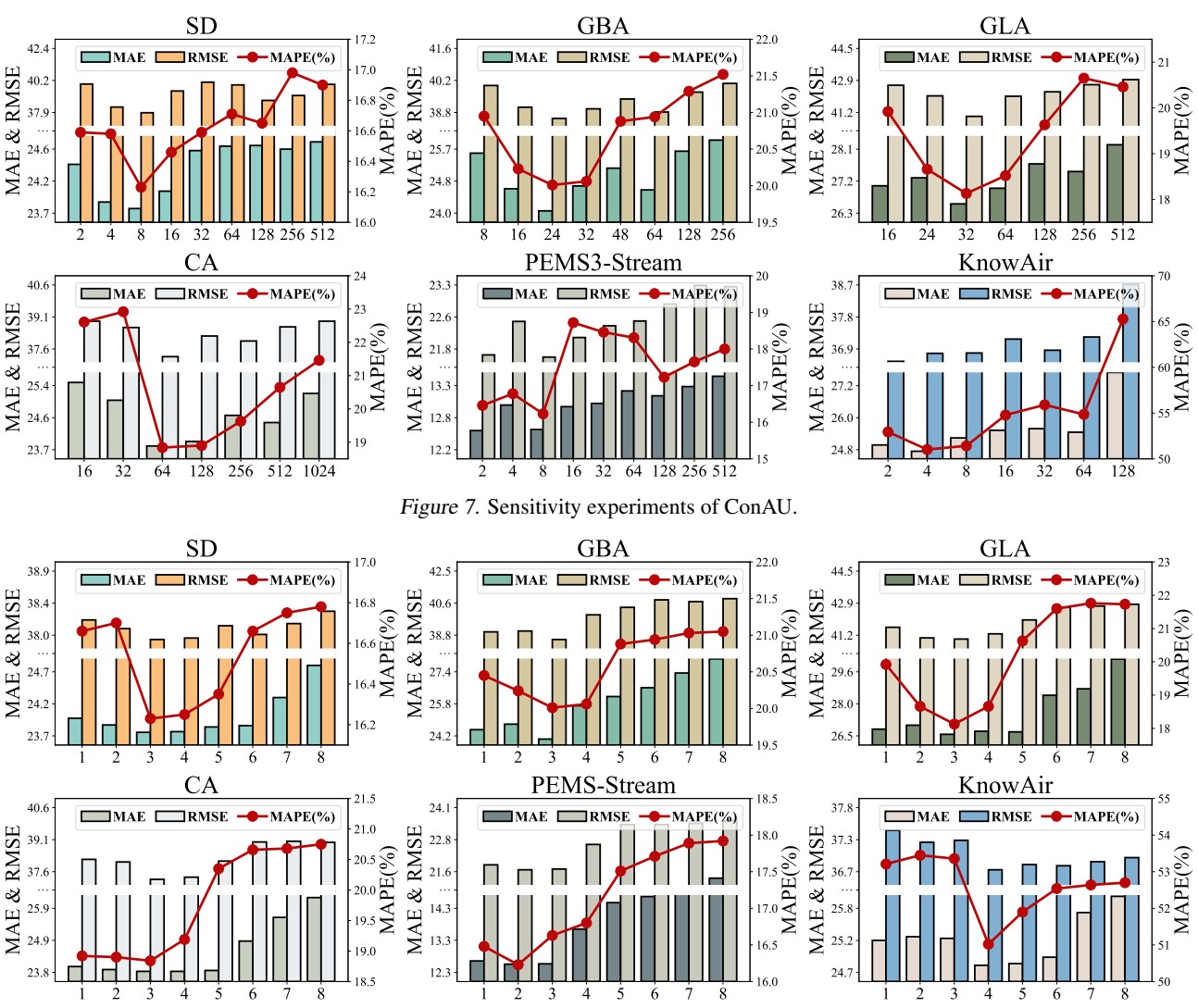

*Figure 7.* Sensitivity experiments of ConAU.

*Figure 8.* Sensitivity experiments of GenPU.

### E.13. Embedding visualization

Using LargeST-SD dataset as example, we visualize the temporal prompt embedding $\mathbf{E}$ in Figure 9 (a), Personalized features $\mathbf{Z}_p$, and contextual features $\mathbf{Z}_c$ in Figure 9 (b). We can see that temporal embeddings unveil essential periodic patterns for OOD scenarios. Both node personalized and context features exhibit strong discriminative capabilities. Context features capture shared node patterns, ensuring resilience to individual node variations. Meanwhile, personalized features enhance the model's ability to tailor predictions for each node effectively.

## F. Discussion

In this section, we discuss the limitations of our work as our future work:

- Validating the Broad Impact of STOP. The spatial interaction module integrated within the STOP framework is inherently generic, suggesting its potential for broader applicability. In upcoming research, we will propose replacing the graph convolutional networks utilized by other spatio-temporal backbones with the spatial interaction module to validate its effectiveness across various contexts. This initiative will help us better understand the potential value and applicability of the STOP module in a wide range of application domains.

- Exploring a Wider Range of OOD Scenarios. Current OOD problems are typically defined within the confines of

*Table 17.* Contribution of prediction components in different scenarios

| Commpent | SD | | KnowAir | |
|---|---|---|---|---|
| | S-OOD | T-OOD | S-OOD | T-ODD |
| $\mathbf{Y}_t$ | 64.04 | 41.95 | 72.62 | 36.23 |
| $\mathbf{Y}_s$ | 35.96 | 58.05 | 27.38 | 63.77 |

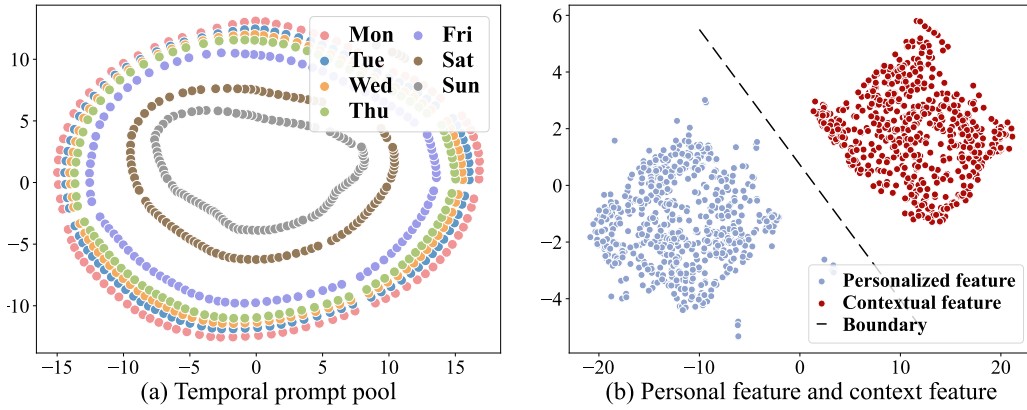

(a) Temporal prompt pool          (b) Personal feature and context feature

*Figure 9.* Visual case and efficiency study of STOP on LargeST-SD dataset.

single-modal data and single tasks. However, spatio-temporal data exhibits diverse modalities and varied tasks. We believe that an improved spatio-temporal OOD handler should be capable of addressing challenges such as cross-task and cross-modal processing, areas that have not been thoroughly explored in the spatio-temporal domain.

- Integrating Large Language Models for zero-shot learning. In OOD settings, accurately predicting new nodes poses a significant challenge, as these nodes have not been encountered by the model during training—commonly referred to as the zero-shot challenge. Large language models excel in this context, as their representational capabilities, developed from extensive training on massive datasets, can enhance a model's zero-shot learning ability (Liu et al., 2025a;c;b). While this has been successfully demonstrated in the time series community, it remains relatively unexplored within the spatio-temporal domain. In future work, we plan to integrate large language models into the STOP framework to further enhance its scalability for predicting new nodes.

