# OpenReview forum: "Robust Spatio-Temporal Centralized Interaction for OOD Learning"
_ICML.cc/2025/Conference — ICML 2025 poster_

### Official Review · Reviewer_5Pwy · 2025-02-25

**Overall Recommendation:** 3

**Summary:**

This authors introduce a Spatio-Temporal OOD Processor, a framework for out-of-distribution learning in spatiotemporal graph convolutional networks. As stated, the traditional methods rely on node to node message interactions, which degrade performance in OOD settings. The proposed method addresses such limitations by using a centralized messaging mechanism with Context-Aware Units and Generalized Perturbation Units -- replacing direct node interactions. A spatiotemporal optimization strategy is used to expose models to diverse training environments and this argued to improve generalization. They evaluate on six datasets.

**Claims And Evidence:**

Node-to-node messaging in STGNNs is sensitive to spatiotemporal shifts and this reduces generalization: Empirical results show that removing direct node-to-node interactions improves robustness in OOD scenarios

The centralized messaging mechanism improves model generalization and robustness: The proposed method higher performance over 14 baselines across 6 datasets.

Message perturbation via GenPU forces models to learn invariant spatiotemporal knowledge: Models trained with GenPU outperform those trained without perturbations in both generalization and inductive learning.

The DRO-based training strategy improves adaptability to unseen environments: The proposed method outperforms ERM methods under spatiotemporal shifts.

The claims and evidences appear to be clear and convincing.

**Essential References Not Discussed:**

Related Work, Section 5, does not appear to be comprehensive enough. Although recent works have been fairly introduces, the prior work leading to the SOTA could be better described

**Experimental Designs Or Analyses:**

Experimental setup and results appear to be well-designed and executed. Ablation studies fairly confirm that replacing node-to-node messaging with ConAUbased centralized messaging improves robustness. Also, removing GenPU results in degraded performance. This tells about its role in improving feature extraction. The proposed method outperforms conventional methods by up to 17.01% in generalization and 18.44% in inductive learning. The proposed method appears to be efficient in large-scale datasets.

This work could also benefit from a discussion on the trade-offs between computational efficiency and model interpretability. Also comparison with meta-learning approaches for generalization could add further depth.

Some question marks down here:

- How does the proposed method perform in traditional IID settings compared to its OOD benefits?

- Could you speak to the computational trade-offs of using centralized messaging versus node-to-node interactions?

- How well does the proposed method scale in real-time applications with large spatiotemporal datasets?

- Could the proposed method be applied to non-spatiotemporal domains such as sequential decision-making or reinforcement learning? Would the proposed modules be of particular use for such use cases?

**Methods And Evaluation Criteria:**

Evaluated on six spatiotemporal datasets and compared against 14 baselines. The evaluation metrics are MAE, RMSE, MAPE. And the experimental settings include multiple OOD settings, including temporal shifts, structural shifts, and rapid expansion scenarios. All these design choices appear to be legit.

**Other Comments Or Suggestions:**

Re context:

- Investigating the impact of the proposed method on long-term temporal dependencies.

- Analyzing the proposed methods interpretability and decision-making process for better model transparency.

- Briefly discussing potential applications in climate modeling, urban planning, and environmental monitoring.

Re writing:

- I do not think a full stop or comma is needed after formulations. This could be revised.

**Other Strengths And Weaknesses:**

- The paper appears to be well-written in terms of language. However, reading is made difficult by the frequent introduction of acronyms. Keeping track of them creates a significant cognitive load, making it harder to follow the content. Simplifying this would greatly enhance readability.

**Questions For Authors:**

No questions apart from those raised in my comments in other sections.

**Relation To Broader Scientific Literature:**

This work builds on prior research in spatiotemporal learning, graph neural networks, and OOD generalization. However, some inconsistencies exist: 1- the study assumes that centralized messaging resolves OOD sensitivity, but it does not thoroughly analyze how different types of spatiotemporal shifts impact performance. 2- the role of message perturbation and DRO in non-spatiotemporal settings is not explored and this effectively limits applicability.

**Theoretical Claims:**

The centralized messaging mechanism reduces complexity to O(KNdh) -- claimed to be more computationally efficient than traditional self-attention methods.

The proposed method's optimization function follows a distributionally robust paradigm and this is claimed to lead to better generalization compared to traditional empirical risk minimization.

---

> ### Author Rebuttal · Authors · 2025-04-01
>
> Thank you very much for your review. We provide figures and tables at an anonymous link ( https://anonymous.4open.science/r/reb_/5Pwy.pdf ). **The "#" before a title means that complete tables in the section are available in the anonymous link.**
>
> > \# Meta-learning
>
> We introduce meta-learning based spatiotemporal models, and results on SD dataset show that our model outperforms these models.
>
> |Model|Ours|MegaCRN|MGT|
> |:-:|:-:|:-:|:-:|
> |MAE|**23.67**|25.47|26.72|
>
> > \# IID
>
> Using SD dataset as an example, the table below reports the average MAE. This demonstrates that our model still achieves competitive performance under the IID setting.
>
> |Model|Our|STONE|CaST|D $^2$ STGNN|STGCN|
> |:-:|:-:|:-:|:-:|:-:|:-:|
> |MAE|**23.59**|24.98|30.17|24.46|25.44|
>
> > Computational Complexity
>
> As demonstrated in Theorem 1, our centralized messaging has higher efficiency. While the computational complexity of node-to-node interaction is O( $N^2$ d), our method reduces it to O(PNd), where P is much smaller than N. For example, P can be as small as 8 for a graph with 716 nodes.
>
> > Large Spatiotemporal Dataset
>
> As noted in line 309, in Section E.3, we evaluate the performance on large datasets. Below, we summarize key results from the CA dataset for easy reference. These results confirm the strong effectiveness of our model for large-scale data.
>
> |Model|Ours|CaST|BigST|GWNet|STGCN|
> |:-:|:-:|:-:|:-:|:-:|:-:|
> |MAE|**32.86**|41.26|39.59|37.63|40.64|
>
> > \# Non-spatiotemporal Domain
>
> We regret that, due to significant conceptual and methodological differences between the spatiotemporal domain we focus on and the reinforcement learning and sequential decision-making areas of interest to you, we cannot explore these unrelated areas within the limited time available.
>
> To address your concerns, we extended our evaluation to discrete graph OOD learning. Following the setup in [1], the table below on Collab-0.8 dataset shows our method performs well in non-spatiotemporal domains. "w/o DRO" and "w/o Per" indicate the removal of the DRO and message perturbation mechanisms, respectively, underscoring their positive effects in non-spatiotemporal setting.
>
> |Method|Ours|w/o DRO|w/o Per|DIDA|DySAT|VREx|
> |:-:|:-:|:-:|:-:|:-:|:-:|:-:|
> |AUC|**77.66**|75.62|73.45|72.59|62.19|62.21|
>
> [1] Zhang Z, Wang X, et al. Out-of-distribution generalized dynamic graph neural network with disentangled intervention and invariance promotion. arXiv preprint.
>
> > Different Types of Shifts
>
> In the third paragraph of the introduction and in Figure 1, we analyze the effects of different shifts. Temporal shifts lead to inaccurate node descriptions, with representation errors propagating through node-to-node messaging and impacting other nodes. spatial shifts disrupt established message passing paths, preventing the model from following trained pathways and degrading its graph representation performance. **Appendix section E.5 evaluates our method’s ability to address these shifts**. To address your concern, we will refine the third paragraph to better emphasize the distinct impacts of each shift.
>
> > Related Work
>
> Sorry for any confusion. Due to space constraints, **we discuss related works to spatiotemporal OOD learning in detail in Appendix Section A**. In the new version, ICML permits an additional page, which will allow us to incorporate this discussion.
>
> > \# Long-term temporal dependencies
>
> Following the setup in long-term time series analysis, we use the Knowair dataset as an example, employing 96 time steps to predict 720 time steps. The averaged results below demonstrate that our proposed model effectively handles long-term temporal dependencies.
>
> |Model|Ours|CaST|BigST|GWNet|STGCN|
> |:-:|:-:|:-:|:-:|:-:|:-:|
> |MAE|**24.89**|26.41|27.76|26.35|28.77|
>
> > Decision-making Process
>
> **Although model interpretability—an active research area—is not the primary focus of this paper, which emphasizes spatiotemporal OOD generalization**—we analyze the prediction decision-making process in Section E.14 using Shapley values, a standard tool for evaluating component contributions. Our model integrates temporal and spatial components to generate the final output. Under spatial shifts, the temporal component dominates, while under temporal shifts, the spatial component takes precedence. Section E.15 further improves interpretability through embedding visualization. In future, we will explore enhancing interpretability by incorporating insights from sequential decision-making or reinforcement learning.
>
> > Potential Applications
>
> Spatiotemporal prediction, to which our method belongs, plays an active role in various downstream applications. For example, traffic forecasting can help urban planners optimize city designs, while meteorology forecasting can guide the public in understanding future climate and environmental conditions.
>
> > Presentation
>
> We will use the full terms instead of acronyms, except for the name of the model STOP. Moreover, we will remove symbols following formulas.

---

> > ### Comment · Reviewer_5Pwy · 2025-04-08
> >
> > My comments are appropriately addressed. I appreciate the explanations and improvements and decide to increase the score.

---

> > > ### Author Response · Authors · 2025-04-09
> > >
> > > Dear reviewer 5Pwy,
> > >
> > > We are deeply grateful for your thoughtful evaluation and for awarding our work with an increased score. Your constructive feedback and recognition are of great value to us, and we are truly honored that you recognize our efforts.
> > >
> > > Sincerely,
> > >
> > > The Authors

---

### Official Review · Reviewer_objW · 2025-03-05

**Overall Recommendation:** 4

**Summary:**

The authors introduce a novel spatio-temporal interaction mechanism called STOP, which is tailored to enhance the sensitivity of the conventional node-to-node messaging mechanism favored by existing models to spatiotemporal shift. Key elements of STOP include the centralized message mechanism, the message perturbation mechanism, and an optimization objective rooted in distributed robust optimization. Through a series of comprehensive experiments, the results effectively showcase the efficacy and competitiveness of STOP.

**Claims And Evidence:**

The authors have elucidated a strong motivation: existing spatio-temporal interaction mechanisms serve as on trigger for the sensitivity of current models to spatio-temporal changes. They have substantiated this motivation through a series of experiments, accompanied by relevant citations.

**Essential References Not Discussed:**

The efforts made by the authors to cover a comprehensive range of related work are commendable. However, to strengthen the literature review, I suggest two improvements: (1) elaborate on the advancements and novelty of the proposed method, and (2) discuss advancements in OOD research in other domains [1- 2].

Ref:
[1] Yang J, Zhou K, Li Y, et al. Generalized out-of-distribution detection: A survey[J]. International Journal of Computer Vision, 2024, 132(12): 5635-5662.
[2] Liu J, Shen Z, He Y, et al. Towards out-of-distribution generalization: A survey[J]. arXiv preprint arXiv:2108.13624, 2021.

**Experimental Designs Or Analyses:**

I reviewed the experiments in the paper.  It includes comparison experiments, double/multiple ablation studies, hyperparameter experiments, efficiency experiments, etc., thoroughly evaluated the proposed method.

(1). Authors are expected to add predictive visualization cases to intuitively assess the model's predictive performance.

(2). Add more discussion on the zero-shot performance of the model.

(3). If we allow fine-tuning using a few data with new features, the authors are requested to compare the model's few-shot learning capabilities.

**Methods And Evaluation Criteria:**

The proposed framework integrates three synergistic technical innovations—centralized messaging,  message perturbation, and distributionally robust optimization—which collectively provide a comprehensive solution to the spatiotemporal shift challenge.

**Other Comments Or Suggestions:**

None.

**Other Strengths And Weaknesses:**

Overall, the paper demonstrates high quality: it presents a convincing motivation, introduces an innovative framework, maintains a well-structured presentation, and conducts very comprehensive experimental evaluations.

However, I have several concerns:

1.	Expand the discussion on continual learning methods and general OOD techniques.

2.	Include an evaluation of the model's effectiveness in few-shotsettings.

3.	Provide visualized prediction cases to illustrate the model's effectiveness.

**Questions For Authors:**

What potential improvements can be made to the STOP framework?

Is the use of GCN essential for spatiotemporal prediction?

**Relation To Broader Scientific Literature:**

The spatiotemporal OOD task that the authors focus on represents a new and emerging research direction in the field of spatiotemporal prediction. Their critical insights into existing message-passing mechanisms are fresh perspectives in this field.

**Theoretical Claims:**

This paper leverages the Distributed Robust Optimization theory to underpin the sophistication of the model. This framework is well-established.

---

> ### Author Rebuttal · Authors · 2025-04-01
>
> Thank you sincerely for your appreciation and thoughtful comments regarding the paper. Your feedback is invaluable in enhancing the quality of the manuscript.
>
> > **W1. Visualization Cases**
>
> Due to the rebuttal policy restrictions of ICML, we were only able to include some prediction visualization cases in an anonymous link (https://anonymous.4open.science/r/_reb/objW.pdf). Meanwhile, in the submitted manuscript, Figure 9 visualizes the time embeddings, personalized embeddings, and contextual embeddings used in our model, providing evidence for the effectiveness of the proposed embedding techniques.
>
> > **W2. Zero-shot Performance**
>
> In fact, the zero-shot performance includes the results for newly added nodes, as shown in Table 4 of our submitted manuscript. These nodes had no observed data during training and were excluded from the training process. Despite this, our model demonstrates impressive zero-shot performance. This is largely due to its ability to extract shared contextual features from spatiotemporal data, which can be leveraged by newly added nodes to achieve effective representations. For convenience, we have included the zero-shot performance results for SD below.
>
> |||||
> |:-:|:-:|:-:|:-:|
> ||MAE|RMSE|MAPE|
> |GWNet|29.01|44.18|22.82|
> |STID|25.40|39.53|18.66|
> |BigST|25.22|39.22|18.02|
> |D $^2$ STGNN|25.85|39.33|20.07|
> |STAEformer|25.28|39.42|18.41|
> |CaST|28.83|44.10|22.93|
> |STONE|25.06|39.15|18.12|
> |Ours|22.74|36.09|16.82|
>
> > **W3. Few-shot Performance**
>
> To address your question, we use the SD dataset as an example. Following the experiment setup in the paper, when the testing environment changes, we fine-tune the model using the first week of the test set to evaluate its few-shot performance. The results are shown in the table below. We find that with minimal fine-tuning data, our model quickly adapts to new spatiotemporal patterns, demonstrating strong generalization. This is largely due to the lightweight architecture and effective use of contextual features.
>
> |||||
> |:-:|:-:|:-:|:-:|
> | |MAE|RMSE|MAPE|
> |STGCN|25.38| 38.77|20.61|
> |GWNet|23.15 | 35.53|18.11|
> |STID|25.47 |39.65|18.87|
> |BigST|25.55| 39.61|18.81|
> |D $^2$ STGNN| 22.09|34.82|16.87|
> |STAEformer| 26.22|40.34|19.15|
> |CaST| 28.40| 43.66|20.70 |
> |STONE| 23.08| 34.88| 17.55|
> |Ours| 20.34| 32.64| 14.62|
>
> > **W4. Related Work**
>
> In Section A.3, we summarize recent advances in continual learning. While these methods address dynamic spatiotemporal data, they typically require extensive new data for fine-tuning, implicitly assuming independent and identically distributed (i.i.d.) data.
>
> In the updated paper, we discuss progress in out-of-distribution (OOD) learning. Traditional machine learning assumes training and testing data share the same distribution, but real-world applications often face distribution shifts, leading to significant performance drops post-deployment [1]. This has driven growing interest in OOD learning, with methods categorized into unsupervised representation learning, supervised model learning, and optimization-based approaches [2-3]. These techniques leverage causal and invariant learning to extract generalizable knowledge from latent test distributions.
>
> Recently, OOD learning in graph learning has gained attention. For example, DisenGCN [4] disentangles informative factors in graph data, assigning them to distinct parts of vector representations. However, these methods struggle with spatiotemporal OOD problems due to their inability to capture complex, heterogeneous spatiotemporal correlations.
>
> Our key contribution is a novel spatiotemporal interaction mechanism that integrates centralized message-passing, message perturbation, and a distributionally robust optimization (DRO) objective, addressing these limitations effectively.
>
> Reference:
>
> [1] Yang, Jingkang, et al. "Generalized out-of-distribution detection: A survey." ICCV 2024.
>
> [2] Liu, Jiashuo, et al. "Towards out-of-distribution generalization: A survey." arXiv preprint arXiv:2108.13624 (2021).
>
> [3] Kaddour, Jean, et al. "Causal machine learning: A survey and open problems." arXiv preprint arXiv:2206.15475 (2022).
>
> [4] Ma, Jianxin, et al. "Disentangled graph convolutional networks." ICML 2019.
>
> > **Q1. Potential Improvement**
>
> In Section F, we discuss potential improvements to the model, including enhancing its performance by integrating large language models and implementing more advanced perturbation mechanisms, among others.
>
> > **Q2. GCN**
>
> Whether GCNs are essential for spatiotemporal forecasting remains debated. However, research consistently shows that GCN-based models outperform those relying solely on temporal dependencies, owing to their ability to introduce inductive biases among nodes. In general OOD scenarios, the dense message-passing mechanism of GCNs exhibits sensitivity to distributional shifts. To address this, we propose a centralized interaction mechanism as an alternative, enhancing the robustness of spatiotemporal interactions.

---

> > ### Comment · Reviewer_objW · 2025-04-09
> >
> > The response has addressed my concerns, so I have decided to raise my score.

---

> > > ### Author Response · Authors · 2025-04-09
> > >
> > > Dear reviewer objW,
> > >
> > > We are deeply grateful for your thoughtful evaluation and for awarding our work. Your constructive feedback and recognition are of great value to us, and we are truly honored that you recognize our efforts.
> > >
> > > Sincerely,
> > >
> > > The Authors.

---

### Official Review · Reviewer_riin · 2025-03-14

**Overall Recommendation:** 4

**Summary:**

The spatiotemporal messaging mechanism utilized in STGCN exhibits inherent sensitivity to spatiotemporal variations. To address these limitations, the authors introduce a centralized messaging architecture integrated with a message perturbation mechanism and DRO optimization. Extensive experimental evaluations were conducted across six benchmark datasets, encompassing comparative analyses with 14 baseline methods. The results indicate that the proposed model achieves performance improvements of up to 17.01% compared to existing approaches.

**Claims And Evidence:**

Yes, the authors focus on the sensitivity of the messaging mechanisms in existing STGNNs to spatio-temporal shifts. They substantiate this by conducting comparative experiments and ablation study involving multiple SOTA STGNNs across diverse OOD scenarios.

**Essential References Not Discussed:**

I believe that much of the work related to the focus of the authors has already been discussed in this paper, primarily in the appendix. I suggest further consolidating the existing progress in the main body of the text. Additionally, I still encourage the authors to discuss general OOD learning research.

**Experimental Designs Or Analyses:**

The experimental section demonstrates rigorous methodology and thorough validation procedures. The study presents a systematic performance evaluation comparing 14 baseline models across six distinct datasets. A comprehensive set of ablation studies has been conducted to quantitatively assess the contribution of individual components. Furthermore, the authors provide detailed analyses including component-wise ablation studies, computational efficiency assessments, and extensive robustness evaluations.

To further strengthen the study, it would be valuable to consider additional experimental investigations, particularly: (1) comparative analyses under more challenging operational scenarios, and (2) detailed assessments of memory efficiency and resource utilization. These supplementary evaluations could provide deeper insights into the model's practical applicability and scalability.

**Methods And Evaluation Criteria:**

Yes, in response to the sensitivity of existing spatiotemporal node-to-node messaging mechanism of STNNs, this study proposes a novel spatiotemporal interaction mechanism that integrates message perturbation and Distributionally Robust Optimization strategies. These strategies are tailored to address specific challenges, and the design motivation is both rational and logically consistent.

**Other Comments Or Suggestions:**

1.	' within Appendix A' in 639 line seems redundant.
2.	Some abbreviations of proper nouns appear repeatedly.
3.	In Table 20, D2STGNN should be written D$^2$STGNN.

**Other Strengths And Weaknesses:**

Strengths:

The authors have identified a critical limitation in existing spatiotemporal messaging mechanisms, specifically their sensitivity to out-of-distribution (OOD) scenarios, which constitutes a significant and well-justified research motivation.

The technical contributions present a novel centralized interaction framework that incorporates message perturbation mechanisms and distributionally robust optimization objectives. The provided theoretical analysis substantiates the proposed model's advantages and robustness properties.

The experimental evaluation demonstrates comprehensive methodology, including extensive comparative analyses across multiple datasets, systematic ablation studies, and detailed efficiency assessments. These empirical results provide substantial evidence supporting the model's performance improvements.

Weaknesses and Recommendations:

The paper would benefit from a broader discussion on general graph OOD learning algorithms, which could provide valuable context and highlight the proposed method's position within the broader research landscape.

The current experimental setup limits dynamic nodes to 30% of the total nodes. To further validate the model's robustness, it is recommended to conduct additional evaluations under more challenging conditions, such as scenarios where 70% of nodes exhibit dynamic behavior.

While computational efficiency has been partially addressed, a more comprehensive evaluation should include memory usage metrics and total training time. These additional metrics would provide a more complete assessment of the model's practical applicability and scalability.

**Questions For Authors:**

Please refer to weaknesses 1-3
Q.1 In Table 3, why does MLP-based STID achieve better prediction performance than Transformer-based D$^2$STGNN?
Q.2 Does STOP include a unique design to address temporal shifts?

**Relation To Broader Scientific Literature:**

While previous research has empirically demonstrated the sensitivity of spatiotemporal prediction models in OOD scenarios, the authors have made a novel contribution by identifying and analyzing the fundamental source of this vulnerability stemming from the spatiotemporal messaging mechanism they adhere to.

**Theoretical Claims:**

Yes, the application of Distributionally Robust Optimization theory in this study theoretically demonstrates that the proposed model exhibits superior generalization capabilities compared to conventional models.

---

> ### Author Rebuttal · Authors · 2025-04-01
>
> We deeply appreciate your valuable time and effort, which are crucial for improving the quality of our manuscript.
>
> > **W1. General Graph OOD Learning Discussion**
>
> In the field of graph representation learning, researchers have focused on modifying model architectures to enhance the representation of invariant knowledge, thereby improving generalization performance for OOD problems [1-2]. For instance, GAUG [3] improves downstream training and inference by incorporating an edge prediction module to modify the input graph. DisenGCN [4] emphasizes learning disentangled representations by separating distinct and informative factors from graph data and assigning them to different parts of the decomposed vector representations. OOD-GNN [5] introduces a nonlinear graph representation decorrelation method, leveraging random Fourier features to eliminate statistical dependencies between causal and non-causal graph representations generated by the graph encoder. However, these models fall short in capturing the complex heterogeneous spatiotemporal correlations inherent in spatiotemporal data, resulting in suboptimal performance for spatiotemporal OOD tasks. In this paper, we propose a novel spatiotemporal interaction module to enhance the robustness of models against spatiotemporal shifts. We will add this discussion to the next version.
>
> > **W2. More challenging conditions**
>
> In Section E.6 of the appendix, we discuss performance comparisons in scenarios where the graph topology grows rapidly. To further evaluate robustness under extreme conditions, we use the SD dataset as an example, where the size of the test graph topology is nine times larger than that of the training graph. The performance results are shown in the figure below.
> ||  | ||
> |------------|-------|------------|-------|
> || MAE| RMSE |MAPE|
> | STGCN| 36.45 | 55.89| 28.01 |
> | GWNet| 37.22 | 56.62| 33.12 |
> | STNorm| 29.79 | 47.71| 24.51 |
> | BigST| 29.72 | 48.55| 21.09 |
> | D $^2$ STGNN | 37.46 |56.37| 29.63 |
> | STAEformer| 28.94 |45.84| 29.40 |
> | CaST| 30.20|47.09| 22.66 |
> | STONE| 36.19|56.71| 30.68 |
> | Ours| **26.73** |**42.35**|**19.38**|
>
> > **W3. Comprehensive Efficiency Analysis**
>
> We sincerely apologize for any lack of clarity in our explanation. We report comprehensive efficiency metrics for several SOTA models on the SD dataset, including simultaneous training time, total training time, inference time, memory usage under optimal performance conditions, and average MAE performance. We find that our proposed method achieves competitive efficiency compared to the SOTA model D $^2$ STGNN. This is attributed to our model's reliance on a lightweight MLP architecture.
>
> | | Average MAE | Train (s/epoch) | Inference (s) | Total (h) | Memory (MB) |
> |:-:|:-:|:-:|:-:|:-:|:-:|
> | STGCN| 25.79| 82.6|7.4| 2.50| 3,783|
> | GWNet| 28.21| 125.3| 14.3| 3.53| 6,871|
> | STNorm | 26.51| 78.2| 7.0 | 2.36| 2,755|
> | BigST| 26.26| 68.3| 7.3| 2.10| 2,791|
> | CaST| 29.84| 82.1| 6.2| 1.52| 3,255|
> | STAEformer | 26.20| 443.7| 36.8 | 12.96| 32,667|
> | D $^2$ STGNN | 25.43| 1216.4| 58.9| 33.65| 50,171|
> | STONE | 25.50 | 199.8 | 16.6 | 5.21| 12,349|
> | Ours | 23.67  | 59.2 | 6.8| 1.54| 3,652|
>
> > **Presentation Problem**
>
> We sincerely appreciate your valuable suggestions. In the new version, we will optimize the structure of the paper and correct typographical errors to further improve the overall quality and readability of the manuscript.
>
> > **Q1. MLP- vs Transformer- based Models**
>
> STID outperforms D $^2$ STGNN in some experimental results, potentially because STID leverages various embedding techniques to capture prior information, which is beneficial for enhancing the model's generalization capability. On the other hand, D2STGNN has a more complex parameter structure, making it prone to overfitting in the training environment. This overfitting can lead to a decline in its ability to generalize to unseen scenarios.
>
> > **Q2. Temporal shifts**
>
> We tackle temporal shifts using two key techniques: (1) Temporal Embedding Technology : By encoding day-of-week and timestamp-of-day information, this approach captures multi-level periodic patterns in spatiotemporal data, strengthening the model's ability to represent stable temporal structures. (2) Decoupling Mechanism : We employ time series decomposition to model seasonal and trend components separately. The stable characteristics of periodic and seasonal patterns enhance the model's robustness against temporal shifts.
>
> Reference:
>
> [1] Park, Hyeonjin, et al. "Metropolis-hastings data augmentation for graph neural networks." NeurIPS 2021.
>
> [2] Wu, Ying-Xin, et al. "Discovering invariant rationales for graph neural networks." ICLR 2022.
>
> [3] Zhao, Tong, et al. "Data augmentation for graph neural networks." AAAI 2021.
>
> [4] Ma, Jianxin, et al. "Disentangled graph convolutional networks." ICML 2019.
>
> [5] Li, Haoyang, et al. "Ood-gnn: Out-of-distribution generalized graph neural network." TKDE 2022.

---

> > ### Comment · Reviewer_riin · 2025-04-03
> >
> > Thanks for the authors' detailed responses. All my concerns have been satisfactorily addressed, and I lean to vote for acceptance.

---

> > > ### Author Response · Authors · 2025-04-09
> > >
> > > Dear reviewer riin,
> > >
> > > Thank you very much for your time, effort, and valuable insights in reviewing our manuscript. Your guidance has been instrumental in helping us refine and improve our work.
> > >
> > > Sincerely,
> > >
> > > The Authors.

---

### Official Review · Reviewer_tJb2 · 2025-03-15

**Overall Recommendation:** 4

**Summary:**

The paper introduces a spatiotemporal interaction mechanism called STOP. STOP includes centralized message passing mechanisms with message perturbation and DRO to enhance stability for spatiotemporal sifts. Through extensive experiments, STOP's robustness to spatiotemporal shifts is effectively demonstrated.

**Claims And Evidence:**

The author identified the limitations of existing methods through experimental observations and used this as motivation for the paper.

**Essential References Not Discussed:**

Related work has been compared or discussed.

**Experimental Designs Or Analyses:**

The authors compared the performance of STOP in various OOD scenarios, including rich baselines and datasets, encompassing multiple common metrics in spatiotemporal learning. They also conducted ablation studies, hyperparameter experiments, and other supplementary experiments to demonstrate the model's effectiveness.

**Methods And Evaluation Criteria:**

The three strategies proposed by the authors to enhance robustness to spatiotemporal shifts are reasonable. The authors have also discussed the roles of each strategy in the method section.

**Other Comments Or Suggestions:**

For efficiency comparison experiments, it is recommended to present the data more intuitively using tables.

**Other Strengths And Weaknesses:**

Strengths of the paper：

-	The centralized node interaction mechanism introduced in this paper is innovative and shows promise as a robust learner for spatial features.

-	The comprehensive OOD benchmark evaluation framework utilized in the experiments is praiseworthy, as it covers multiple datasets, rich baselines, and comparisons across different OOD scenarios.

-	The paper's assertion that the traditional node-to-node messaging mechanism is fragile is a new finding.

Weaknesses of the paper：
-	Efficiency comparisons are available on two datasets in paper, and there is a lack of presentation of efficiency comparisons for large-scale dataset.

-	More discussion on baseline experiment details and . As far as I know, some models have parameters coupled with graph size, such as GWNet and AGCRN. How authors handle adaptation to OOD settings?

**Questions For Authors:**

-	What is the role of the spatio-temporal prompt in STOP for spatiotemporal shift?
-	How efficient is STOP on the CA dataset?
-	If traditional GCN is limited in OOD scenarios, then there is a natural question: Is GCN really necessary?
-	How is the masking matrix of the message interference mechanism sampled from the distribution?

**Relation To Broader Scientific Literature:**

The centralized interaction mechanism proposed by the author can broaden the architectural perspectives of existing models in the field.

**Theoretical Claims:**

The formulas in the paper are correct and align with the descriptions provided. The use of DRO theory in the paper can explain its advancements.

---

> ### Author Rebuttal · Authors · 2025-04-01
>
> Thank you very much for your appreciation and comments on the paper. Your comments are crucial to improving the quality of the manuscript.
>
> > **W1 & Q2. Efficiency Analysis**
>
> We sincerely apologize for any misunderstanding caused. Below, we report the efficiency comparison between the proposed method and several advanced models on the CA dataset. The efficiency metrics include memory usage, training time per epoch, total training time, and the amount of memory used. We can find that our model achieves competitive performance while maintaining high efficiency
>
> | CA     | MAE | Train (s/epoch) | Inference time (s)   | Total (h) | Memory (MB)  |
> |:--------:|:--------:|:-----------------:|:-------:|:-----------:|:---------:|
> | STGCN  | 40.64  | 951.2| 219.54 | 23.80| 26,395  |
> | GWNet  | 37.63  | 2279.6| 332.92 | 69.66| 34,103  |
> | STNorm | 40.77  | 360.8| 69.40  | 11.13| 29,275  |
> | BigST  | 39.59  | 328.4| 82.91  | 10.78| 16,283  |
> | CaST   | 41.26  | 577.3| 153.72 | 17.09| 19,515  |
> | Ours   | 32.86 | 290.2| 70.58 | 7.13| 15,243  |
>
> > **W2. OOD Setting**
>
> This issue is addressed in Appendix E.1. For models such as GWNet and AGCRN, they employ an adaptive graph learning enhancement technique, which generates meaningful node representations for each node but also causes the parameter scale to become coupled with the size of the graph structure. To address this, we have removed this technique.
>
> > **Q1. Spatio-temporal prompt**
>
> This technique uses embeddings to encode the day-of-week and timestamp-of-day information. These prior knowledge capture the periodic patterns in spatiotemporal data, which are relatively stable and help improve the model's generalization ability against spatiotemporal shifts.
>
> > **Q3. Traditional GCN**
>
> Whether GCN is necessary in the field of spatiotemporal prediction has long been a topic of debate. However, many studies in the spatiotemporal domain have consistently shown that GCN-based models outperform those that only consider temporal dependencies, as they introduce an inductive bias between nodes. In spatiotemporal OOD tasks, the dense message-passing mechanism of GCNs demonstrates sensitivity to spatiotemporal shifts. Therefore, we focus on enhancing the robustness of spatiotemporal interactions and propose a centralized interaction mechanism as an alternative solution.
>
> > **Q4. Message Interference Mechanism**
>
> In lines 188 to 197, we describe the random sampling process for the stochastic masking matrix. Specifically,  we first create $M$ learnable probability vectors and normalize it, with the result denoted as {$ p_1^\prime, p_2^\prime,\dots, p_M^\prime$}, where ${p}_i^\prime \in \mathbb{R}^N$ with $i \in$ {$1, 2, \cdots, M$} means $i$-th probability vector. Then, for $i$-th probability vector, we can establish a binomial distribution, which is denoted as $\mathcal{M}\left({p}_i^\prime;s\right)$. Using this distribution, we can sample a masking indices $\widetilde{g}_i \sim \mathcal{M} \left({p}_i^\prime;s \right) \in$ {0,1}$^N$, where $s\in\left(0,N\right)$ indicates the number of sample hits (i.e. the number of values equal to 1 in $\widetilde{\boldsymbol{g}}_i$). Finally, we create $K$ replicas of $\widetilde{\boldsymbol{g}}_i$. As a result, we can obtain a mask matrix with log operation, and the output is denoted as $\mathbf{G}_i=\log\left(\left[\widetilde{\boldsymbol{g}}_i,\widetilde{\boldsymbol{g}}_i,\dots,\widetilde{\boldsymbol{g}}_i\right]\right)\in\lbrace-\infty,0\rbrace^{K\times N}$ and $\mathbf{G}_i$ is used to interfere with the message process. Similarly, we ultimately perform multiple rounds of perturbations on the message-passing mechanism using $M$ interference matrices. To avoid further confusion, we will refine this section to improve clarity and ensure a more precise presentation.

---

> > ### Comment · Reviewer_tJb2 · 2025-04-02
> >
> > Most of my concerns are appropriately addressed. I choose to increase the score.

---

> > > ### Author Response · Authors · 2025-04-09
> > >
> > > Dear Reviewer tJb2,
> > >
> > > We are deeply grateful for your thoughtful evaluation and for awarding our work with an increased score. Your constructive feedback and recognition hold immense value for us, and we are truly honored by your acknowledgment of our efforts.
> > >
> > > Sincerely,
> > >
> > > The Authors

---

### Decision · Program_Chairs · 2025-05-01

**Decision:**

Accept (poster)

**Comment:**

This paper presents a well-motivated and technically sound method for spatiotemporal OOD learning. The proposed framework addresses a fundamental limitation in existing STGNNs by replacing node-to-node messaging with centralized interactions enhanced by message perturbation and robust optimization. The authors provide extensive evidence of performance gains, theoretical justification, and real-world efficiency while showing strong rebuttal responsiveness. While minor limitations remain like presentation clarity and broader framing, the overall contribution is novel and experimental validated.

In the rebuttal period, the authors actively engaged with reviewers, offering detailed clarifications and new results—such as meta-learning comparisons, long-term forecasting, visualization cases, and cross-domain validation—which directly addressed reviewer concerns and led to multiple score increases.

The reviewers largely converge toward acceptance, and the AC recommend acceptance as well.